ecology, behaviour, physiology

ecological niche, foundation species, functional habitat, individual preference, microclimate, spinifex

**Author for correspondence:**
Kristian J. Bell
e-mail: bellkr@deakin.edu.au

# Predators, prey or temperature? Mechanisms driving niche use of a foundation plant species by specialist lizards

Kristian J. Bell[1], Tim S. Doherty[1,2] and Don A. Driscoll[1]

[1]Centre for Integrative Ecology, School of Life and Environmental Sciences, Deakin University, Geelong, Victoria, Australia
[2]School of Life and Environmental Sciences, University of Sydney, Sydney, New South Wales, Australia

KJB, 0000-0002-1857-6257; TSD, 0000-0001-7745-0251; DAD, 0000-0002-1560-5235

Foundation species interact strongly with other species to profoundly influence communities, such as by providing food, refuge from predators or beneficial microclimates. We tested relative support for these mechanisms using spinifex grass (*Triodia* spp.), which is a foundation species of arid Australia that provides habitat for diverse lizard communities. We first compared the attributes of live and dead spinifex, bare ground and a structurally similar plant (*Lomandra effusa*), and then tested the relative strength of association of two spinifex specialist lizard species (*Ctenophorus spinodomus* and *Ctenotus atlas*) with spinifex using a mesocosm experiment. Temperatures were coolest within spinifex compared to bare ground and *Lomandra*. Invertebrate abundance and the threat of predation were indistinguishable between treatments, suggesting temperature attenuation may be a more important driver. Overall, the dragon *C. spinodomus* preferred live over dead spinifex, while the skink *C. atlas* preferred dead spinifex, particularly at warmer air temperatures. However, both species displayed individual variability in their use of available microhabitats, with some individuals rarely using spinifex. Our results provide an example of temperature attenuation by a foundation species driving niche use by ectothermic animals.

## 1. Introduction

Niche theory underpins almost every aspect of ecology [1], with the concept describing the complex and dynamic interaction of organisms with environmental variables, constrained by resource limitations, competition and predation [2]. In essence, animals must consume adequate food and water, avoid predation, tolerate or avoid abiotic stresses and reproduce [3]. Trophic interactions may be modulated by resources or predators, reflecting bottom-up versus top-down controls, respectively [4]. Some taxonomic groups can be more strongly influenced by specific aspects of niche space. For example, the majority of lizard communities are believed to be partitioned according to spatial niche requirements [5], whereas snakes generally partition through diet [6]. Furthermore, factors defining niches can depend on the spatial scale [7]. Teasing out which aspects of the environment define niches is therefore an essential pursuit in ecology.

Foundation species create complex habitats that are fundamental to the structure, function and resilience of ecosystems [8]. Complex structures provide diverse microhabitat availability and increase food resources, refuges and available niche space [9], thus fostering high biodiversity [10,11]. Coral reefs, for example, promote species coexistence through the amelioration of physical stress and the creation of fine-scale, complex matrices in which smaller organisms can find refuge [8,12]. Foundation species can also have a disproportionate influence on abiotic processes such as hydrology, nutrient cycling, humidity and

temperature [13,14]. Altered thermal conditions can in turn affect the behaviour of temperature-sensitive animals and ultimately lead to shifts in community structure and ecosystem function [15]. Indeed, the attenuation of temperatures by foundation species, often with favourable outcomes for associated biota, has been recorded across numerous systems, including marine (macroalgae [16]), temperate forest (eastern hemlock [17]) and alpine environments (cushion plant [18]).

In addition to these well-established abiotic impacts, foundation species can also affect biotic processes such as species interactions and resource acquisition [19,20], often through their impact on the physical structure of the environment. For example, declines in a foundation species of tree (*Tsuga canadensis*) led to a more open canopy, which increased decomposition rates, and altered ant and plant assemblages in North American forests [21]. Similarly, in arid Australia, the structure of hummock-forming *Triodia* grasses promotes a high diversity of termites and ants, and a high diversity of lizards that consume them [5,22]. The inter-connected nature of vegetation structure and biotic and abiotic factors makes it challenging to determine precisely how foundation species influence other organisms. For example, the physical form of *Triodia* hummocks may alter temperature regimes and substrate properties, facilitating its use by burrowing animals, which in turn further alter habitat properties.

The impact of foundation species can be sufficiently strong that their loss or decline may precipitate fundamental changes to communities [23] and abruptly restructure ecosystems [24,25]. Such habitat-forming organisms are often disproportionately impacted by disturbance [23], and other organisms can become increasingly dependent on foundation species as environmental stress increases [26,27]. Reductions in habitat complexity can favour generalist, invasive or disturbance-tolerant species, thus leading to community homogenization [10]. Impacts on biodiversity may therefore be particularly strong where foundation species are in decline, due to both the large number of interacting species and the loss of potential niche space. As such, an improved understanding of the functional role of foundation species will enable us to better understand wider ecosystem functions and better predict responses of species and systems to global change.

The arid zone spinifex grasslands of Australia contain the highest lizard species diversity and richness on Earth [28] and numerous studies identify the close relationship of lizards with spinifex [28–30]. However, the mechanisms behind this association are not understood. In this study, we investigated the relative support for three mechanisms potentially driving animal use of a foundation species by asking: (i) does a foundation plant species provide temperature regimes, food resources or protection from predators that are distinctive from other available microhabitats, and (ii) for animals reported to be associated with that plant, does microhabitat choice reinforce the preferential use of the foundation species and its underlying mechanisms. To address these questions, we used the close association of lizard species from two families (Agamidae and Scincidae) and the foundation plant species spinifex (*Triodia scariosa*) as a model system.

## 2. Methods

### (a) Study site and species

Spinifex (*Triodia* spp.) is a hummock-forming grass that is the dominant ground cover over approximately one-quarter of the Australian continent [31]. Spinifex is a foundation species due to its effect on faunal diversity and ecosystem processes [32]. The densely matted interior stems and rigid, needle-like leaf tips of spinifex (figure 1) provide an important structure for many birds [33,34], reptiles [30,35], mammals [36,37] and invertebrates [38,39]. Our study area was the Nombinnie and Round Hill Nature Reserve complex covering an area of approximately 70 000 hectares in New South Wales, Australia (−33.03, 146.11). Survey locations were dominated by spinifex grass (*T. scariosa*) interspersed with mallee eucalypt trees. For field experiments, we used two lizard species (*Ctenophorus spinodomus* (Agamidae) and *Ctenotus atlas* (Scincidae); figure 1) that are strongly associated with spinifex grass and are spinifex specialists [29,40,41].

Our experimental design involved four microhabitat types as treatments: live spinifex, dead spinifex, bare ground, and a sympatric, non-spinifex plant. The herb *Lomandra effusa* (hereafter *Lomandra*) was chosen as the non-spinifex plant as it most closely resembles the structure, dimensions and ecology of spinifex within the study area (electronic supplementary material, appendix figure S1). Like spinifex, *Lomandra* is a native, perennial plant with rigid leaves that grow to 30–50 cm, although *Lomandra* forms a dense tussock rather than the open hummock typical of mature *T. scariosa*. The leaves of *Lomandra* are not as tightly interwoven, nor as sharp and needle-like as *T. scariosa*. We included a dead spinifex treatment as it offers a similar internal structure and spiky exterior to live spinifex and so probably offers similar protection from predators. However, it may differ in temperature or prey availability and thus may help tease apart the relative strengths of competing mechanisms.

### (b) Temperature regimes

We deployed 120 iButton temperature loggers (Maxim Integrated Products, Sunnyvale, CA, USA) arranged in blocks of four across the study area (mean distance between blocks = 17.9 m, mean distance within blocks = 4.0 m). Within each block, a logger was placed in a live spinifex plant, a dead spinifex plant, a *Lomandra* plant and an open patch of bare ground (greater than 1 m from plant cover), giving 30 replicates of these blocks. Temperatures at ground level were measured every 6 min for 9 days, with data for half the sites recorded from 22 October 2019 and the remaining sites from 18 November 2019. iButtons were positioned flush to the ground and as deep as possible within the core of plants without damaging the structural integrity of the plant. We recorded the size of the individual plants by measuring the length and height of each clump, excluding inflorescences.

To test for differences in the thermal regime between microhabitats, we fitted a linear mixed model with microhabitat type as a fixed effect and block and individual plant (repeated measure) as random effects. These analyses and all others that follow were performed using the *glmmTMB* and *lsmeans* packages in R [42,43]. To examine the effects of physiologically challenging temperatures (hereafter referred to as 'extreme') for our study species, we used an upper threshold of 45°C and a lower threshold of 9.7°C. These temperatures represent the approximate critical thermal maximum ($CT_{max}$) of *C. spinodomus* and two congeneric species of *Ctenotus* (*Ctenotus regius* and *Ctenotus uber*) with overlapping distributions to *C. atlas*, and the average critical thermal minimum ($CT_{min}$) temperature of *C. regius* and *C. uber*, respectively [29,44]. We were unable to find suitable data on the $CT_{min}$ for *C. spinodomus*, nor other sympatric *Ctenophorus* spp., thus we used the same value as for *C. atlas*. We counted both the number of days, and the duration within each day in which values either exceeded the upper limit or dropped below the lower limit within each treatment over the 18-day survey period. For reporting purposes, we assumed a 6 min duration each time an extreme temperature was recorded, which reflects the frequency at which temperature was logged. We used a hurdle model with a Poisson distribution to

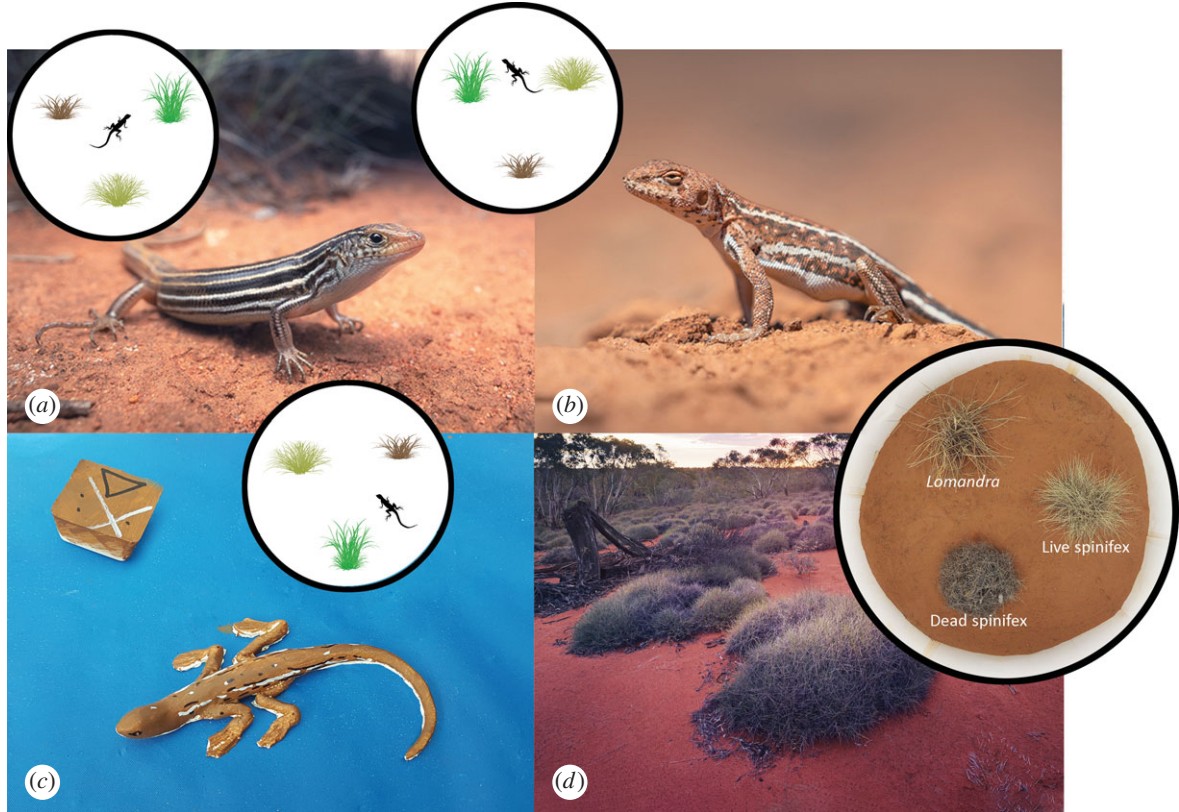

**Figure 1.** (*a*) Southern mallee ctenotus (*Ctenotus atlas*) and (*b*) mallee dragon (*Ctenophorus spinodomus*) from our study site. Lizard model used in predation experiment to mimic *Ctenophorus spinodomus* and procedural control cube (*c*) and typical mature spinifex clumps and open mallee habitat (*d*). Left overlay: study design showing spatial configuration of the three enclosures in relation to each other, with rotation of treatment types within each enclosure. Right overlay: example enclosure for mesocosm experiment as viewed from monitoring camera. (Online version in colour.)

determine (i) if the number of days where extreme temperatures were recorded varied between microhabitat types and (ii) if the duration of extreme temperature events varied between microhabitat types. We specified survey day as a random effect. Subsequent pairwise comparisons were made using the Tukey method. To investigate if differences in temperature could be attributed to spinifex clump size, we fitted a linear mixed model, with clump size, clump height and spinifex type (live/dead) as interacting fixed effects, and block and individual plant (repeated measure) as random effects.

## (c) Invertebrate sampling

We sampled invertebrates using the same design as for temperature measurements (see electronic supplementary material, appendix for full sampling details). We tested for differences in invertebrate abundances between microhabitats using generalized linear mixed models. We used three different measures of invertebrate abundance: total abundance of all invertebrates and abundance of an assemblage reflecting the dietary composition of each of the two study species. *Ctenotus atlas* has a varied diet consisting predominantly of Hymenoptera, Araneae, Coleoptera, Isoptera, Blattodia and Orthoptera [45] (see electronic supplementary material, appendix table S1), while *C. spinodomus* feed almost exclusively on ants (Formicidae) [29]. We used negative binomial models because they provided an appropriate fit for the data (assessed via the *DHARMa* package [46,47]). We included a random effect of the block to account for the nested spatial structure of the sampling design and conducted post hoc pairwise comparisons using the Tukey method. To investigate if differences in invertebrate abundance could be attributed to spinifex clump size, we fitted a linear model with clump width, clump height and clump type (live or dead) as interacting fixed factors, and total invertebrate abundance as the dependent variable.

## (d) Predator surveys

We assessed predation pressure by placing lizard models in each of the four microhabitats at 28 sites (minimum inter-site distance = 800 m), with models within sites separated by at least 5 m each (total = 112 models, mean within-block distance = 18.3 m). The models were designed to mimic the general shape, size and colour pattern of *Ctenophorus spinodomus* (figure 1) and were left in the field for 7 consecutive days between November and December 2019. We monitored models using motion-sensing cameras and also inspected models for any signs of displacement, breakage, wear or other disturbance consistent with a predatory attack to quantify any interactions missed by the cameras (see electronic supplementary material, appendix for further details).

## (e) Mesocosm experiment

We conducted a mesocosm experiment to provide two elements of supporting information, namely (i) the strength of selection for live spinifex relative to other microhabitats and (ii) whether microhabitat preferences change as temperatures change, thus supporting the temperature hypothesis. We used pitfall trapping and active searching to catch 40 *Ctenophorus spinodomus* and 24 *Ctenotus atlas*. Lizards were individually introduced to one of three, identical, semi-natural enclosures (2.2 m diameter), comprising a bare sand substrate and an open canopy (figure 1). In each enclosure, a similarly sized (approximately 40 cm crown width) live spinifex, dead spinifex and live *Lomandra* plant were placed an even distance apart and kept clear of each other and the sides of the enclosure (figure 1). We retained as much of the root clump of plants as possible, but removed excess retained soil by hand prior to positioning within mesocosms. We selected dead spinifex clumps that still retained the overall shape, height and structure of live clumps to minimize any differences in clump volume. The position of each plant type was switched between the three enclosures to control for any

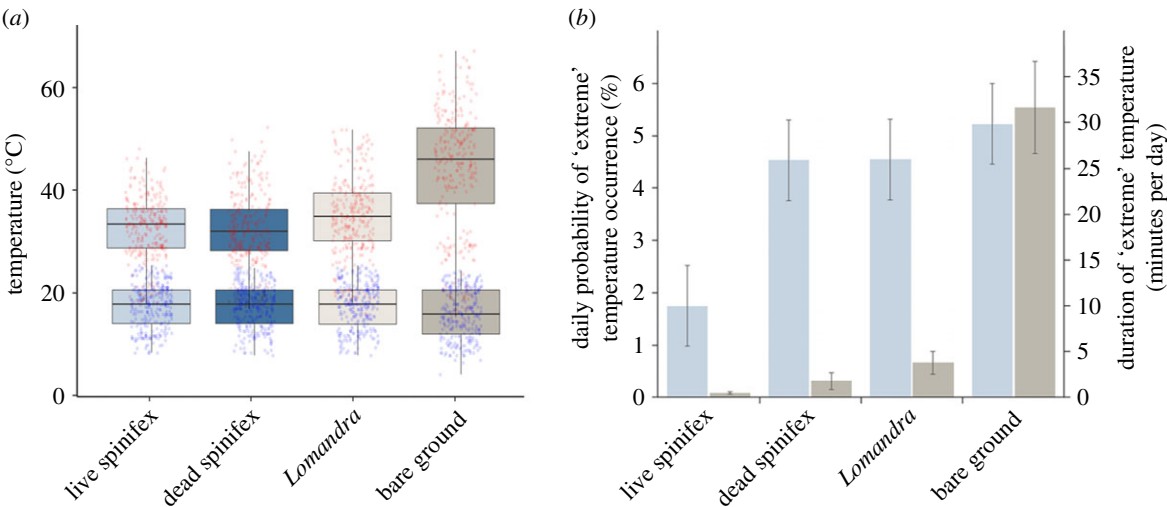

**Figure 2.** (*a*) Maximum and minimum daily temperatures within treatments. Whiskers represent values within 1.5 times above or below the 75th and 25th interquartile range, respectively. Red and blue dots represent raw values for maximum and minimum temperatures respectively. (*b*) Probability of occurrence (blue bars) and duration (brown bars) of ecologically 'extreme' temperatures per day over the 18 days of temperature logger deployment. (Online version in colour.)

potential effects of slope or shading from the enclosure walls. We also placed iButton temperature loggers within treatments inside enclosures to assess the effects of transplanting and the influence of the mesocosms on irradiation exposure.

Once placed in enclosures, lizards were left undisturbed for 3–5 h, during which time a 12MP YI action camera attached to an overhead beam and looking directly down at the enclosure recorded a single-colour image every 5 s. At the end of the trial, the lizards were marked on both the ventral and dorsal side with a non-toxic marker to ensure no individual was tested again, and returned to their point of capture. Trials were conducted in three week-long sessions in October and November 2019, and February 2020.

Images were analysed using ExifPro (v. 2.1), with each frame assigned a tag for whether the lizard was within/underneath the crown of one of the plants, out in the open, or if its location was unknown. Habitat selection was tested using the number of observations of each animal within each treatment type as the dependent variable in a generalized linear mixed model with a negative binomial distribution. We specified enclosure identity [1–3], time of trial (am/pm), microhabitat, sex, snout-vent length (SVL) and average temperature as fixed effects, as well as interactions between microhabitat type and temperature, microhabitat type and sex, and microhabitat type and SVL. We also specified the total number of observations per trial as an offset [48]. Individual lizards were specified as random effects to account for non-independence of observations (e.g. more counts of a lizard in live spinifex mean fewer counts in other treatments). We also included the proportional use of each treatment by lizards in the previous trial, up to a maximum of 24 h prior, as a covariate to control for the influence of olfactory cues left by previously tested lizards on microhabitat selection of subsequent lizards. Where no lizard was tested within the previous time period, we used the average use of that treatment type by the species as the value.

## 3. Results

### (a) Thermal regimes

Temperatures ranged from 3.1°C to 68.9°C and differed significantly between treatments ($\chi^2 = 88.355$, $p < 0.001$). The bare ground treatment had the highest mean daily temperatures and greatest fluctuations (27.3°C mean daily temperature, 95% CI 26.5–28.1). The coolest temperatures were recorded within

dead spinifex (24.0°C, 23.2–24.8) and live spinifex (24.1°C, 23.3–24.9), and *Lomandra* had intermediate temperatures and stability (25.0, 24.2–25.8). Pairwise comparisons showed that bare ground had significantly higher temperatures than all other treatments ($p < 0.001$) and live and dead spinifex significantly lower temperatures than *Lomandra* ($p = 0.023$ and 0.010, respectively), but temperatures between live and dead spinifex were not statistically different from one another ($p = 0.748$). Analysis of mean daily maximum and minimum temperatures revealed similar patterns, with bare ground recording significantly cooler minimum daily temperatures (15.9°C, 15.3–16.6, $p < 0.001$) and significantly warmer maximum daily temperatures (43.2°C, 41.6–44.8, $p < 0.001$) than all other treatments (figure 2*a*). Live spinifex and dead spinifex were statistically indistinguishable from one another ($p = 0.823$) for both minimum (17.5°C, 16.8–18.1) and maximum (live = 31.8°C, 30.2–33.5; dead = 31.6°C, 30.0–33.3) daily temperatures. *Lomandra* only differed statistically from spinifex at maximum daily temperatures (34.1°C, 32.5–35.7, $p < 0.001$).

Spinifex clump sizes within treatments spanned a height range from 11 to 98 cm (mean live = 44.3 cm, s.d. = 11.8; mean dead = 32.0 cm, s.d. = 11.9), and widths ranged from 45 to 199 cm (mean live = 71.6 cm, s.d. = 24.6; mean dead = 78 cm, s.e. = 33.9). Live spinifex had the lowest number of days with 'extreme' temperatures (5 of 18 days), followed by dead spinifex and *Lomandra* (13 of 18 days) and bare ground (15 of 18 days; figure 2*b*). All pairwise comparisons were statistically significant ($p < 0.05$). Live and dead spinifex experienced the smallest average duration of extreme temperatures (1.2 and 1.8 min per site per day, respectively), followed by *Lomandra* (3.8) and bare ground (31.7). Differences in the duration of extremes were statistically significant between all pairs of treatment types ($p < 0.001$; figure 2*b*). Neither the height (model coefficient = −0.163, $p = 0.557$), width (0.479, $p = 0.222$), nor interaction of height and width (0.595, $p = 0.213$) of spinifex clumps, were significant in explaining differences in average daily temperature.

### (b) Invertebrate abundance

We sampled 16 089 individual invertebrates, with ants comprising 60% of all captures. Fewer invertebrates were

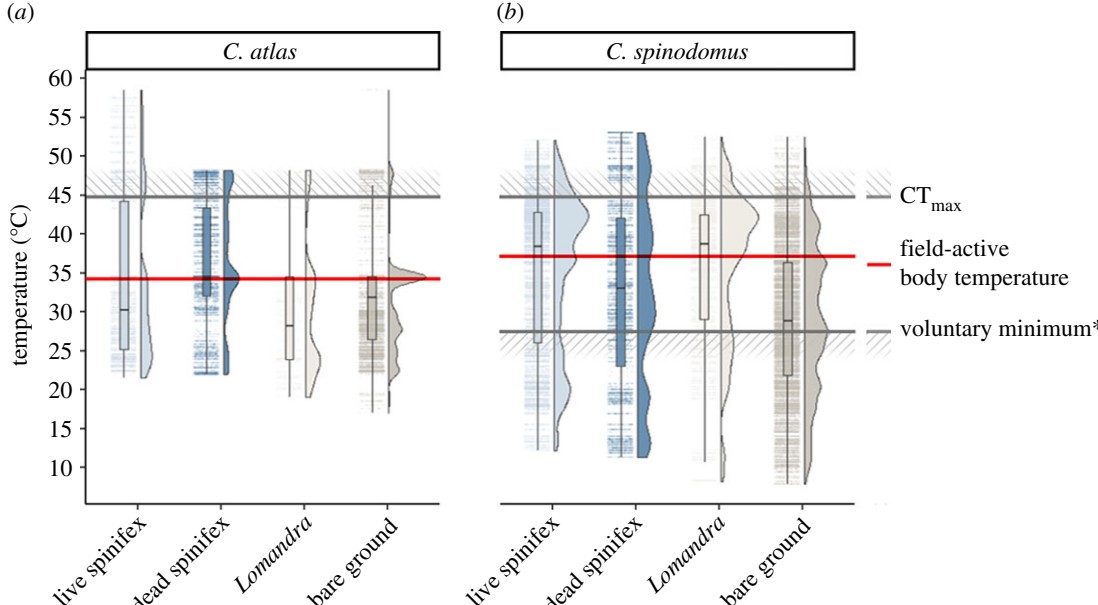

**Figure 3.** Variation in microhabitat use with temperature for two lizard species during mesocosm experiments. Individual temperature observations for lizards within each microhabitat are represented by dots and data density curves. Upper diagonal hatched lines represent the approximate $CT_{max}$ of *C. atlas* and *C. spinodomus* (45°C). The horizontal red lines represent the field-active body temperatures of each species [29,44,49]. Lower diagonal hatched line represents the voluntary minimum for *C. spinodomus* (26.8°C; *equivalent data not available for *C. atlas*). The voluntary minimum is an activity threshold temperature, where lizards voluntarily emerge from refuges and become active. (Online version in colour.)

trapped in live spinifex than in bare ground (model coefficient = 0.20, $p = 0.046$), though this difference was not significant once we applied the Tukey adjustment (electronic supplementary material, appendix figure S2). There were no other differences in total invertebrate abundance between treatments. Similarly, we found no difference in ant abundance (reflecting *C. spinodomus* diet) across treatment types ($\chi^2 = 6.81$, $p = 0.078$), nor in the abundance of the assemblage reflecting *C. atlas* diet ($\chi^2 = 1.15$, $p = 0.764$). The size of spinifex clumps did not affect the abundance of invertebrates (height $-5.070$, $p = 0.062$; width $-3.752$, $p = 0.080$; interaction 1.176, $p = 0.068$).

## (c) Predator surveys

We reviewed 4700 h of surveillance across all lizard models, recording a total of 24 species and 131 individual animal observations. However, only eight encounters with models (where an animal either tasted, touched or closely inspected the model) were recorded across all treatments, of which only a single clear 'predation' event was recorded—a brown songlark *Megalurus cruralis* attacking a model in the bare ground treatment. Given the limited number of observations, the differences were not statistically meaningful. The only mammalian predator recorded was a single cat *Felis catus* (near *Lomandra*), with no foxes *Vulpes vulpes* detected throughout the surveillance period. During pitfall trapping (50 traps deployed for 18 days), captures of potential lizard predators included five varanid lizards (*Varanus gouldii*), one elapid snake (*Suta dwyeri*), two pygopods (*Pygopus schraderi*) and 18 small mammals (1 × *Mus musculus*, 12 × *Ningaui yvonneae*, 5 × *Sminthopsis murina*).

## (d) Mesocosm experiment

A total of 147 861 individual observations across both lizard species were recorded, covering approximately 225 h of footage.

There were 5468 observations (9.9% of 55 086 total observations) for the skink *C. atlas* where temperatures exceeded the $CT_{max}$ threshold (greater than 45°C). Similarly, 7,732 observations (8.3% of 92 775 total observations) exceeded the $CT_{max}$ for the dragon *C. spinodomus*. A further 13 857 observations (14.9% of total) occurred below the voluntary minimum (26.8°C) for *C. spinodomus* (figure 3).

Preferences between microhabitats were highly variable among individuals (electronic supplementary material, appendix figures S3 and S4). Overall, *C. atlas* was most commonly observed on bare ground (model coefficient = 6.98, 95% CI = 6.54–7.42), followed by dead and live spinifex (6.06, 5.65–6.48; 5.69, 5.24–6.15), and *Lomandra* (5.25, 4.68–5.82; figure 3). Use of bare ground was significantly higher than all other treatment types ($p < 0.001$), and use of dead spinifex was higher than *Lomandra* (0.813, $p = 0.011$). Temperature was important and had a significant interaction with treatment ($\chi^2 = 15.696$, $p = 0.001$), with *C. atlas* increasing use of dead spinifex relative to live spinifex as temperatures increased (0.257, $p < 0.001$). The use of bare ground also decreased with increasing temperatures, but to a lesser degree than live spinifex (0.115, $p = 0.018$; figure 4). The remaining variables had no relationship with microhabitat preferences (sex: $\chi^2 = 0.380$, $p = 0.944$; SVL: $\chi^2 = 4.708$, $p = 0.195$; time of trial: $\chi^2 = 1.355$, $p = 0.508$; enclosure: $\chi^2 = 0.331$, $p = 0.847$; preferences of the previously tested lizard: $\chi^2 = 2.513$, $p = 0.113$).

*Ctenophorus spinodomus* discriminated between microhabitats, and used bare ground the most (6.97, 6.55–7.40), followed by live spinifex (6.12, 5.67–6.57), *Lomandra* (5.49, 4.93–6.04) and dead spinifex (4.75, 4.08–5.41), though the use of dead spinifex was not statistically distinguishable from *Lomandra* ($-0.738$, $p = 0.060$). There was a weak interaction between temperature and treatment ($\chi^2 = 6.154$, $p = 0.104$), with *C. spinodomus* increasing use of live spinifex compared to bare ground ($-0.072$, $p = 0.037$) and dead spinifex ($-0.106$, $p = 0.030$) at higher temperatures (figure 4). Microhabitat

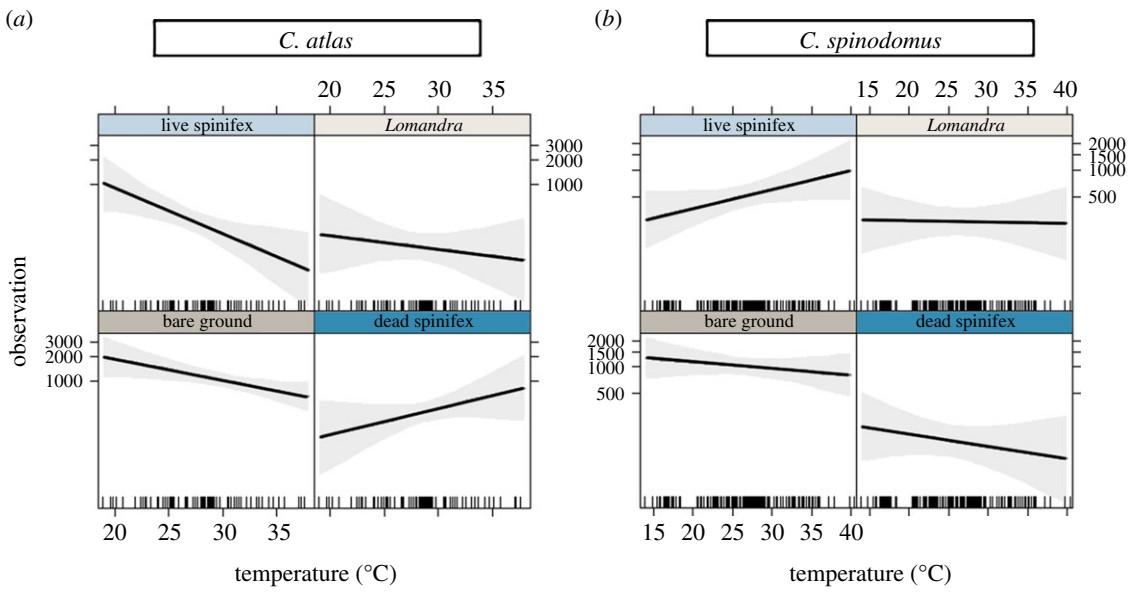

**Figure 4.** Interactions of habitat choice and temperature from the mesocosm experiment. Observations are standardized to account for differences in the total number of observations per trial. (Online version in colour.)

preferences were not influenced by the preferences of previously tested lizards ($\chi^2 = 0.010$, $p = 0.922$), time of trial ($\chi^2 = 0.039$, $p = 0.843$), enclosure ($\chi^2 = 2.060$, $p = 0.357$), sex ($\chi^2 = 4.227$, $p = 0.238$) nor SVL ($\chi^2 = 2.634$, $p = 0.452$).

iButton temperature loggers placed within treatments inside enclosures revealed relative temperature patterns in line with the thermal regime study, with significant differences in temperatures ($F_{3,8} = 26.0$, $p < 0.001$). Pairwise comparisons were also consistent within enclosures compared to the field experiment, though temperatures within live spinifex could no longer be statistically differentiated from *Lomandra* (electronic supplementary material, appendix figure S5).

## 4. Discussion

Foundation species enhance niche diversity and availability, thus supporting a wide range of biota and increasing food web complexity [50]. We tested the functional environment of a foundation species (spinifex grass *T. scariosa*) for two putative spinifex specialist lizard species. We found no evidence of differences in food abundance or predation rates between spinifex and alternative microhabitats, whereas spinifex provided significantly cooler and less extreme temperatures than bare ground or the structurally similar *Lomandra*. Our findings are congruent with other studies showing that temperature may be more important than food or predation for microhabitat selection in lizards [51–53]. Spinifex (either live or dead) was on average 1°C cooler than *Lomandra* and more than 3°C cooler than adjacent bare ground. Spinifex also reduced both the likelihood and duration of extreme temperatures compared to *Lomandra* and bare ground. Interestingly, the size of spinifex clumps did not appear to influence these thermal patterns. As we conducted our temperature surveys in spring, we speculate that patterns of average and extreme temperatures would be more pronounced during summer and winter, when the potential value of *Triodia* as a thermal refuge may be substantially higher. Terrestrial ectotherms have limited physiological thermal tolerance, and asymmetric temperature-fitness curves mean fitness is

depressed more rapidly at body temperatures above optimal levels [54,55]. Thus, the capacity of spinifex to attenuate temperatures may therefore represent an invaluable thermal refuge to arid-dwelling lizards.

The preference of *C. spinodomus* for live spinifex over dead, and the opposite pattern for *C. atlas*, is interesting given that live and dead spinifex provided comparable thermal regimes and prey availability. These preferences also became stronger with increasing temperatures, providing superficial support for the thermoregulation hypothesis. However, *C. atlas* exhibit cooler field-active body and air temperatures (body 34.5°C; air 29.3°C [49]) than *C. spinodomus* (body 36.9°C; air 30.1°C [29]), but dead spinifex recorded more frequent and longer extreme temperatures than live spinifex. It therefore seems unlikely that temperature tolerances are entirely facilitating this effect. The two species have different diets and foraging strategies, with *C. atlas* predominantly foraging on insects within spinifex clumps [49], while *C. spinodomus* forage on ants in open areas adjacent to spinifex [29]. *Ctenotus atlas* frequently climb within spinifex to 30 cm or more [49] and take prey items from the upper tips of spinifex clumps (K.J.B. 2019, personal communication). While speculative, the reduction in height as dead spinifex degrades may facilitate opportunistic ambushing of prey items that land on the periphery of plants by *C. atlas* hidden within the clump, but this clearly needs further study.

A striking and unexpected result of the mesocosm experiment was the wide-ranging use of plant types across individuals of both species (electronic supplementary material, appendix figure S4). There is evidence of individual-level niche specialization across a broad range of taxa [56] and, although we did not set out to test this, our study adds to the growing literature on behavioural variation among individuals. *Ctenophorus spinodomus* exhibited a near-maximal range of use of live spinifex over other vegetation, from almost total avoidance (1.3% use) to exclusive use (100%). Similarly, individual preferences for dead spinifex within *C. atlas* ranged from 3.5% to 90.6%. Although some of the overall variation in plant selection was due to temperature, thermal effects did not appear to strongly influence differences between individuals. That is, lizards trialled at the same time (in separate mesocosms), and therefore under

comparable environmental conditions, often preferred different plant types (for example see *C. atlas* lizards 3 and 17, electronic supplementary material, appendix figure S4). While *C. atlas* and *C. spinodomus* are considered to have a high reliance on spinifex [40,41], previous studies have not directly addressed intraspecific differences. However, the willingness of both *C. spinodomus* and *C. atlas* to use all treatment types (live and dead spinifex, bare ground and *Lomandra*) despite being considered spinifex specialists may have important implications. Habitat specialization often leads to increased vulnerability to disturbance and higher extinction risk [57,58], and individual-level behavioural variation in microhabitat choice may become more important for populations to survive acute site-level disturbances, such as livestock grazing [45,59].

Within our mesocosm experiment both species overwhelmingly spent more time on bare ground than within any single plant. This was probably due to basking, as reptiles often select relatively warm open microhabitat, particularly when temperatures are below thermal optimums [60–62]. However, our ability to interpret this strong preference for open microhabitat is limited by the relatively low proportion of time at which lizards were subject to thermal stress. The preference of lizards for bare ground may also be due to the unsuitability of treatment plants within the mesocosms, with lizards possibly remaining unusually active at more stressful temperatures in search of more appropriate refuges. For example, the unavoidable use of small plants within mesocosms means we did not capture the potentially greater thermal benefits that larger tussocks may provide. Further, the alteration of hydrological function from transplanting the plants may have reduced differences in stomatal conductance, humidity and subsequently temperature differences through evaporation. Even so, relative differences in temperature within the treatment plants were consistent between mesocosms and the field. Besides thermoregulation, it is possible that courtship display or home-site defence, best performed in conspicuous areas, could partially explain preferences for open microhabitat [47,63]. However, sex was not a significant covariate and lizards were tested individually so our results are unlikely to have been influenced by such behaviours.

Habitat structural complexity is an important factor influencing the abundance and trophic dynamics of invertebrates [11,64]. As such, the lack of difference in invertebrate abundance between treatments is somewhat surprising, given *T. scariosa* arguably has a more complex structure (greater quantity of thin, interwoven leaves and stolons), and therefore more effective surface area. Greater structural complexity may facilitate a greater abundance of climbing species, which are less likely to be caught in pitfall traps. However, the invertebrate groups that correspond to the diets of our study lizards were well represented within our data. Pitfall traps may more accurately reflect the degree of activity rather than abundance and can have a highly variable relationship to underlying invertebrate population densities [65]. It is conceivable therefore that the animals moved at a scale that exceeded the spatial variation in treatment plant occurrence. Regardless of catchability issues, the slightly lower average abundances within the two spinifex treatments compared with *Lomandra* and bare ground, and lack of overall differences across all treatments and species-specific dietary groups, suggests food availability is not a strong driver of microhabitat choice in this system.

The paucity of predation events we detected points to either methodological limitations, a temporary scarcity of predators,

or a minor role of predation in determining microhabitat selection for lizards in this system. For example, the use of a larger number of models may have enabled us to record more predatory events and potentially reveal differences between treatments. Further, at the time of the surveys the study area was experiencing a prolonged drought that may have temporarily reduced the abundance of some predators such as non-native foxes and cats. These two predators frequently prey on small lizards in Australia, including both of our study species or closely related species [34,66,67]. Previous work has shown that avian and mammalian predators attacked lizard models in the open more than those in spinifex dominated areas [68], or at the base of spinifex clumps [69]. Given the range of predators that our cameras were unlikely to detect (e.g. reptiles and small mammals), the limitations of using static models and the timing of trials with a prolonged drought, we cannot rule out a role for predation in microhabitat choice. The addition of a perceived predator to our mesocosm experiment, perhaps in the form of scent or models, could help determine whether the use of spinifex differs with predation risk.

Plant structure, microclimate and species interactions may all influence each other, making it hard to isolate specific effects. Given this challenge, a fully factorial design where multiple parameters of vegetation and predatory stimuli are manipulated may be of most assistance in determining the niche mechanisms that shape the use of foundation species [70]. For example, comparing lizard responses to visual and olfactory predatory cues may help determine if spinifex use depends on threat type (e.g. aerial versus ground-based). In addition, extending mesocosm trials through the night may provide greater insight into refuge preferences when daily temperatures are likely to be at their coolest, and behaviour associated with foraging is substantially reduced or eliminated.

We have shown that thermal factors appear to be the dominant mechanism driving the use of a foundation plant species by two lizard species. Despite temperature being the most important mechanism for both lizard species, their use of spinifex differed at different temperatures, suggesting other aspects of their ecophysiology or ecology may also play an important role. Plant communities are being altered across the world [71], which can change the thermal landscape and substantially affect the behaviour of ectotherms [72]. Disruption of thermal niches may be particularly pertinent within the arid zone of Australia, which contains high reptile endemicity and species richness [28]. Understanding the mechanistic links between an organism's environment and its fitness is critical to predicting animal responses to novel conditions, such as those brought about by climate change, species introductions [73] or declines of foundation species [24,74]. While additional work is required to refine our understanding of competing mechanisms, we have illustrated how a foundation species can alter abiotic conditions and influence functional habitat and niche use in lizards.

Ethics. The research was conducted under a scientific licence issued by the Department of Planning, Industry and Environment, NSW (SL102239), and Deakin University Animal Ethics (B04-2019).

Data accessibility. All data required for the analyses presented in this paper are available from the Dryad Digital Repository: https://doi.org/10.5061/dryad.g79cnp5nv [75].

Authors' contributions. All authors conceived and designed the study. K.J.B. collected the data. K.J.B. analysed the data and wrote the paper, with input from all co-authors.

**Competing interests.** We declare we have no competing interests.

**Funding.** This study was generously supported by the Hermon Slade Foundation (HSF 17/12) and Deakin University's Centre for Integrative Ecology. T.S.D. was supported by an Alfred Deakin Postdoctoral Research Fellowship (Deakin University) and a Discovery Early Career Researcher Award (Australian Research Council).

**Acknowledgements.** We acknowledge the Ngiyampaa Wangaaypuwan people as the Traditional Custodians of the land on which this research was conducted. We thank Mal Carnegie (Lake Cowal Conservation Centre) for his generous supply of equipment, Nick and Callum Porch for processing invertebrate data, and Owen Lishmund, Blake Hose, Kim Fidder, Jack Dickson, Mia Gourlay and George Huggett for their tireless work in the field and lab.

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
