## [Peer Review File · Proceedings of the Royal Society B: Biological Sciences]

Review History

RSPB-2020-2633.R0 (Original submission)

Review form: Reviewer 1 (Sean Tomlinson)

Recommendation

Major revision is needed (please make suggestions in comments)

Scientific importance: Is the manuscript an original and important contribution to its field?

Excellent

General interest: Is the paper of sufficient general interest?

Excellent

Quality of the paper: Is the overall quality of the paper suitable?

Marginal

Is the length of the paper justified?

Yes

Should the paper be seen by a specialist statistical reviewer?

No

Do you have any concerns about statistical analyses in this paper? If so, please specify them explicitly in your report.

Yes

It is a condition of publication that authors make their supporting data, code and materials available - either as supplementary material or hosted in an external repository. Please rate, if applicable, the supporting data on the following criteria.

Is it accessible?

Yes

Is it clear?

Yes

Is it adequate?

Yes

Do you have any ethical concerns with this paper?

No

Comments to the Author

Please see attached review document. (See Appendix A)

Review form: Reviewer 2

Recommendation

Accept with minor revision (please list in comments)

Scientific importance: Is the manuscript an original and important contribution to its field?

Good

General interest: Is the paper of sufficient general interest?

Good

Quality of the paper: Is the overall quality of the paper suitable?

Excellent

Is the length of the paper justified?

Yes

Should the paper be seen by a specialist statistical reviewer?

No

Do you have any concerns about statistical analyses in this paper? If so, please specify them explicitly in your report.

No

It is a condition of publication that authors make their supporting data, code and materials available - either as supplementary material or hosted in an external repository. Please rate, if applicable, the supporting data on the following criteria.

Is it accessible?

Yes

Is it clear?

Yes

Is it adequate?

Yes

Do you have any ethical concerns with this paper?

No

Comments to the Author

This paper examines the role of spinifex as a foundation species for two species of dragon, using observations in the wild and mesocosms to tease apart the reasons these dragons prefer spinifex. In general, I thought the study was excellent, and really tried to examine a series of causes in detail. Limitations of the study are thoroughly discussed in the discussion.

Two small areas might warrant further discussion. One is sex (and reproductive condition) – the wide variation among individuals in habitat preference could, in part, be explained by sex or by whether females are gravid, but that was not mentioned here. Perhaps there was no evidence that sex or reproductive condition drove these differences, but if the authors have the data it might be worth mentioning.

In addition, the ‘failure’ of the predation experiment using models could be examined more fully. Typically, very, very large numbers of models, over long periods, are required to detect predation (I’m not sure what 4,700 hours means in terms of risk), so there may just have not been enough models. Looking back at the methods you don’t actually say how many models there were, that might be good to put in the paper although I suppose it might be in the appendix - I think it should appear in the methods.

Given this failure, it might have been nice to see a graph of the different predators you found in the different habitat types (as part of your trapping study, not just on the cameras) – just to get an idea about potential (and also, maybe, perceived) predation in the different habitats. The lizards might respond to expected predation (or perceived predation risk) rather than actual predation. Further experiments in the mesocosms with predators (in cages, or model predators) could be instructive, and might be worth mentioning.

The paper was, in general, well written, but there were a couple of places where the writing could be improved.

Wordy statements, especially ‘considered to be’ appear everywhere. If something is ‘considered to be’ then it just is, unless you are going to show otherwise. So please cut these out

e.g., Lines 91-93 The effect of spinifex on faunal diversity and ecosystem processes has led to it being considered a foundation species. – wordy and awkward sentence – rewrite

Lines 203-210 – small points but: temperatures differed significantly (rather than just differed – they always differ), and the temperature of bare ground was higher than all the other treatments, and the temperature of spinifex was lower... and they could be differentiated from one another, but were not significantly different.

Line 211: it was not the live spinifex that recorded the lowest number of days...

Line 223: You don’t need to say what kind of plots they are, start with “mean daily temperatures” etc

Line 391: Burton’s legless lizard

Decision letter (RSPB-2020-2633.R0)

22-Dec-2020

Dear Mr Bell:

Your manuscript has now been peer reviewed and the reviews have been assessed by an Associate Editor. The reviewers' comments (not including confidential comments to the Editor) and the comments from the Associate Editor are included at the end of this email for your reference. As you will see, the reviewers and the Editors have raised some concerns with your manuscript and we would like to invite you to revise your manuscript to address them.

Research ethics:

Use of animals and field studies:

It is a condition of publication that you make available the data and research materials supporting the results in the article. Please see our Data Sharing Policies (<https://royalsociety.org/journals/authors/author-guidelines/#data>). Datasets should be deposited in an appropriate publicly available repository and details of the associated accession number, link or DOI to the datasets must be included in the Data Accessibility section of the

article (<https://royalsociety.org/journals/ethics-policies/data-sharing-mining/>). Reference(s) to datasets should also be included in the reference list of the article with DOIs (where available).

Please submit a copy of your revised paper within three weeks. If we do not hear from you within this time your manuscript will be rejected. If you are unable to meet this deadline please let us know as soon as possible, as we may be able to grant a short extension.

Best wishes,
Dr Locke Rowe
mailto: proceedingsb@royalsociety.org

Associate Editor

Board Member: 1

Comments to Author:

This is an interesting paper that could be a significant contribution to the field. Nonetheless, the reviewers think the authors did not analyze the data properly to explain their results in the framework of modern thermal biology of ectotherms. The reviewers provide detailed and insightful comments that may help the authors to make a major revision to the paper.

Reviewer(s)' Comments to Author:

Referee: 1

Comments to the Author(s)

Please see attached review document

Referee: 2

Comments to the Author(s)

This paper examines the role of spinifex as a foundation species for two species of dragon, using observations in the wild and mesocosms to tease apart the reasons these dragons prefer spinifex.

In general, I thought the study was excellent, and really tried to examine a series of causes in detail. Limitations of the study are thoroughly discussed in the discussion.

Two small areas might warrant further discussion. One is sex (and reproductive condition) – the wide variation among individuals in habitat preference could, in part, be explained by sex or by whether females are gravid, but that was not mentioned here. Perhaps there was no evidence that sex or reproductive condition drove these differences, but if the authors have the data it might be worth mentioning.

In addition, the ‘failure’ of the predation experiment using models could be examined more fully. Typically, very, very large numbers of models, over long periods, are required to detect predation (I’m not sure what 4,700 hours means in terms of risk), so there may just have not been enough models. Looking back at the methods you don’t actually say how many models there were, that might be good to put in the paper although I suppose it might be in the appendix - I think it should appear in the methods.

Given this failure, it might have been nice to see a graph of the different predators you found in the different habitat types (as part of your trapping study, not just on the cameras) – just to get an idea about potential (and also, maybe, perceived) predation in the different habitats. The lizards might respond to expected predation (or perceived predation risk) rather than actual predation.

Further experiments in the mesocosms with predators (in cages, or model predators) could be instructive, and might be worth mentioning.

The paper was, in general, well written, but there were a couple of places where the writing could be improved.

Wordy statements, especially ‘considered to be’ appear everywhere. If something is ‘considered to be’ then it just is, unless you are going to show otherwise. So please cut these out

e.g., Lines 91-93 The effect of spinifex on faunal diversity and ecosystem processes has led to it being considered a foundation species. – wordy and awkward sentence – rewrite

Lines 203-210 – small points but: temperatures differed significantly (rather than just differed – they always differ), and the temperature of bare ground was higher than all the other treatments, and the temperature of spinifex was lower... and they could be differentiated from one another, but were not significantly different.

Line 211: it was not the live spinifex that recorded the lowest number of days...

Line 223: You don’t need to say what kind of plots they are, start with “mean daily temperatures” etc

Line 391: Burton’s legless lizard

Author's Response to Decision Letter for (RSPB-2020-2633.R0)

See Appendix B.

RSPB-2020-2633.R1 (Revision)

Review form: Reviewer 1 (Sean Tomlinson)

Recommendation

Accept with minor revision (please list in comments)

Scientific importance: Is the manuscript an original and important contribution to its field?

Excellent

General interest: Is the paper of sufficient general interest?

Good

Quality of the paper: Is the overall quality of the paper suitable?

Good

Is the length of the paper justified?

Yes

Should the paper be seen by a specialist statistical reviewer?

No

Do you have any concerns about statistical analyses in this paper? If so, please specify them explicitly in your report.

No

It is a condition of publication that authors make their supporting data, code and materials available - either as supplementary material or hosted in an external repository. Please rate, if applicable, the supporting data on the following criteria.

Is it accessible?

Yes

Is it clear?

Yes

Is it adequate?

Yes

Do you have any ethical concerns with this paper?

No

Comments to the Author

Please see the attached document. (See Appendix C)

Decision letter (RSPB-2020-2633.R1)

01-Mar-2021

Dear Mr Bell

I am pleased to inform you that your Review manuscript RSPB-2020-2633.R1 entitled "Predators, prey or temperature? Mechanisms driving niche use of a foundation plant species by specialist lizards" has been accepted for publication in Proceedings B.

The referee(s) do not recommend any further changes, with the exception of one typo noted by the referee. That typo is:

L348: 'microhabitat. particularly' The full stop should be a comma.

Therefore, please proof-read your manuscript carefully and upload your final files for publication. Because the schedule for publication is very tight, it is a condition of publication that you submit the revised version of your manuscript within 7 days. If you do not think you will be able to meet this date please let me know immediately.

To upload your manuscript, log into <http://mc.manuscriptcentral.com/prsb> and enter your Author Centre, where you will find your manuscript title listed under "Manuscripts with Decisions." Under "Actions," click on "Create a Revision." Your manuscript number has been appended to denote a revision.

You will be unable to make your revisions on the originally submitted version of the manuscript. Instead, upload a new version through your Author Centre.

1) A text file of the manuscript (doc, txt, rtf or tex), including the references, tables (including captions) and figure captions. Please remove any tracked changes from the text before submission. PDF files are not an accepted format for the "Main Document".

2) A separate electronic file of each figure (tiff, EPS or print-quality PDF preferred). The format should be produced directly from original creation package, or original software format. Please note that PowerPoint files are not accepted.

3) Electronic supplementary material: this should be contained in a separate file from the main text and the file name should contain the author's name and journal name, e.g. `authorname_procb_ESM_figures.pdf`

All supplementary materials accompanying an accepted article will be treated as in their final form. They will be published alongside the paper on the journal website and posted on the online figshare repository. Files on figshare will be made available approximately one week before the accompanying article so that the supplementary material can be attributed a unique DOI. Please see: <https://royalsociety.org/journals/authors/author-guidelines/>

4) Data-Sharing and data citation

It is a condition of publication that data supporting your paper are made available. Data should be made available either in the electronic supplementary material or through an appropriate repository. Details of how to access data should be included in your paper. Please see <https://royalsociety.org/journals/ethics-policies/data-sharing-mining/> for more details.

<http://datadryad.org/submit?journalID=RSPB&manu=RSPB-2020-2633.R1> which will take you to your unique entry in the Dryad repository.

Once again, thank you for submitting your manuscript to Proceedings B and I look forward to receiving your final version. If you have any questions at all, please do not hesitate to get in touch.

Sincerely,
Dr Locke Rowe
Editor, Proceedings B
mailto:proceedingsb@royalsociety.org

Reviewer(s)' Comments to Author:

Referee: 1

Comments to the Author(s)
Please see the attached document

Decision letter (RSPB-2020-2633.R2)

05-Mar-2021

Dear Mr Bell

I am pleased to inform you that your manuscript entitled "Predators, prey or temperature? Mechanisms driving niche use of a foundation plant species by specialist lizards" has been accepted for publication in Proceedings B.

Open Access

You are invited to opt for Open Access, making your freely available to all as soon as it is ready for publication under a CC BY licence. Our article processing charge for Open Access is £1700. Corresponding authors from member institutions (<http://royalsocietypublishing.org/site/librarians/allmembers.xhtml>) receive a 25% discount to these charges. For more information please visit <http://royalsocietypublishing.org/open-access>.

Paper charges

Sincerely,
Proceedings B
<mailto:proceedingsb@royalsociety.org>

Appendix A

Kristian Bell, Timothy Doherty and Don Driscoll "Predators, prey or temperature? Mechanisms driving niche use of a foundation plant species by specialist lizards"

Submitted to *Proceedings of the Royal Society B*: RSPB-2020-2633

Bell and colleagues undertook an interesting study consistent with an emerging literature that has re-engaged with the recognition that thermal biology can be a major attribute structuring animal communities and ecosystems and driving species' niche selection. The rationale and design of the study seem solid enough to me (although with a few confounding elements that require consideration), and the question is intrinsically interesting. I find, however, that the authors lack precision in the discussion of some of the concepts. Their treatment of temperature and its effects is a bit shallow, and as such becomes conflated with other interpretations that I don't think they intended, but which are certainly implied when considering the thermal biology and ecophysiology of ectotherms.

What is a little bit problematic is that this shallow understanding of thermal biology and what can influence thermal biology at local scales has apparently bled over into the methods in some ways. Some of the methods seeking to quantify local temperatures are confounded by seasonal timing and by experimental procedures, and do not provide the strongest insight into the thermal niche and preferences of the lizards. In a similar way, the results aren't presented in the most convincing way, given the data that the authors collected, nor the relevant patterns in the data that the authors overlooked. I recognise that some of these are unavoidable, and none of them are deal-breakers. I certainly don't suggest that they invalidate the study, but they do require either re-analysis and better presentation, or specific consideration and specific discussion in light of the objectives of the study.

My impression is that this imprecision and shallowness comes from a naivete of the field that they have engaged with. There are several key references for work such as this that the authors appear to be unaware of, and these references lay out some very specific structure to how thermal biology, energetics and water budgets are interrelated in a broadly ecophysiological paradigm. Reading these earlier reports would also go some way to indicating that the authors' contribution, while of a high standard and interest, isn't quite so unique and novel as they appear to believe that it is. Indeed, several contributions of this sort have been published in *Proc Roy Soc B* within the last 2 years, addressing various related themes around thermal biology. Ideally, I'd expect to see references to:

Tuff, K. T., et al. (2016). "A framework for integrating thermal biology into fragmentation research." *Ecology Letters* DOI: [10.1111/ele.12579](https://doi.org/10.1111/ele.12579).

Garcia, R. A. and S. Clusella-Trullas (2019). "Thermal landscape change as a driver of ectotherm responses to plant invasions." *Proceedings of the Royal Society B* **286**: 20191020.

Angilletta, M. J. J. (2009). *Thermal Adaptation: a Theoretical and Empirical Synthesis*. Oxford, Oxford University Press.

Kearney, M. (2006). "Habitat, environment and niche: what are we modelling?" *Oikos* **115**: 186-191.

Saleeba, K., et al. (2020). "Using biophysical models to improve survey efficiency for cryptic ectotherms." *Journal of Wildlife Management* <https://doi.org/10.1002/jwmg.21890>.

Seebacher, F. and C. E. Franklin (2005). "Physiological mechanisms of thermoregulation in reptiles: a review." *Journal of Comparative Physiology B* **175**: 533-541.

Tomlinson, S. (2019). "The mathematics of thermal sub-optimality: Nonlinear regression approaches to thermal performance in reptile metabolic rates." Journal of Thermal Biology **IN PRESS**.

Tomlinson, S. (2020). "The construction of small-scale, quasi-mechanistic spatial models of insect energetics in habitat restoration: a case study of beetles in Western Australia." Diversity and Distributions **IN PRESS**.

Tomlinson, S., et al. (2017). "Incorporating biophysical ecology into high-resolution restoration targets: insect pollinator habitat suitability models." Restoration Ecology **26**: 338-347.

The authors can feel free to overlook the last three that I authored, because I recognise the pretension of putting my own work forward like this, and they're not directly applicable to the question at hand, but they might provide some insights into where I think these studies are going. The rest of these references I think are probably critical.

I am also kind of left wondering "so what" with this manuscript. The authors opened broadly with a sort of classically ecological gambit on niche theory. The findings are consistent with what most people would expect of reptile niches, at least in the Australian context. Unfortunately, the study is pervaded by the naivete concerning thermal biology generally and the thermal biology of the focal species, so the discussion flails. The talking points and patterns in the data that are apparent and insightful are missed. Instead the authors resort to a traditional ecological approach talking about resource partitioning, predation pressures and coincidental convergent evolution, even going so far as to suggest that the central premise of their study, that foundation species are important in structuring ecosystems, is in fact false. I've made a number of suggestions on how the authors can access the patterns in the data that I think are in there, and which I think are actually informative, but which they have so far overlooked. Once the authors understand this, then they can approach their discussion more rationally, with a recognition of the constraints imposed by the confounders of their experimental design. Maybe then we can see whether there is a more interesting story that might emerge concerning the potential changes to species assemblages driven by the impact of fire regimes, grazing, and ecological restoration on *Triodia*. Given that Australia's arid zone is noted as hosting one of the most speciose and unique reptile communities in the world, what kind of insights are we gaining by understanding the role of thermal biology at high resolution? I'd like to see these ideas explored and discussed.

So to be clear, I love this kind of research re-engaging thermal biology and ecological energetics with broader ecology. It's the kind of stuff that I've devoted large slabs of my time to, and I really think studies like this need to be published. I also think that the broader ecological literature has kind of forgotten how much we actually know about how these mechanisms work, and how much insight can be gained from properly incorporating ecophysiology and thermal biology back into ecology. I think that this study is a really good example of that, but before it's published, I'd like to see it explored to its fullest potential. Despite the apparent extensive criticism in my review, I'm strongly urging the handling editor to recommend a major revision of this manuscript. I firmly believe that there will be something in here that is worth publishing, and of interest to the audience of *Proc. Roy. Soc. B*. It's clearly just the case that this is the work of a PhD student who hasn't captured the appropriate literature to unravel his data properly, and he deserves to be given the best opportunity to do so.

I've included line by line comments. Hopefully they're useful, but equally they're often just talking points, or points of misunderstanding where I feel that an audience needs clarification. There's also a reference list at the end of the review. I don't normally do that, but there were a number of papers

that I thought that the authors might find specifically useful as I worked my way through the manuscript. I hope that this advice is taken in the spirit in which it was intended.

Sincerely,

Dr. Sean Tomlinson
Research Fellow

School of Biological Sciences
The University of Adelaide
SA 5005 AUSTRALIA
Email: sean.tomlinson@adelaide.edu.au
www.adelaide.edu.au
CRICOS provider number 00123M

Kings Park Science Adjunct
Department of Biodiversity, Conservation and Attractions
WA 6005 AUSTRALIA

THE UNIVERSITY
of ADELAIDE

The logo for Kings Park & Botanic Garden, featuring a stylized plant with a red flower.
KINGS PARK | BOLD PARK
& Botanic Garden

Line-by-line comments

Abstract

- L14: “Foundation” species is a term I’ve not encountered before. I get what it is, it’s clearly analogous to what I think of as “keystone” species. I don’t really play in the pure ecology literature much, but is there a specific reason for changing the term? I’m honestly just curious
- L18: “Resources” is a particularly loose term for what you looked at, and somewhat misleading in the context. Temperature is not a resource, it’s a physical attribute of the microclimate. Similarly, predation risk isn’t exactly a “resource” either. Given that one of the major drivers of microhabitat selection by ectotherms is thermo-energetic, when you say “resources”, I immediately expect something measured in Joules. Maybe “attributes” or something similar is a better term
- L20: Maybe identify your species here. It’s a bit of a cliffhanger waiting for 2 lines to see what they actually were 😊
- L25: Dead spinifex with increasing temperatures? Spinifex doesn’t have temperatures. I think you mean that it preferred dead spinifex, especially at warmer air temperatures
- L26: What three plant types? Live spinifex, dead spinifex and *Lomandra*, or spinifex, *Lomandra* and bare ground? Or live spinifex, dead spinifex, *Lomandra* or bare ground?
- L26: Variation in individual preference is an interesting finding. It’s impossible to unpack in an Abstract, but I think it’s too simplistic to say “spinifex specialists should use spinifex” when you’ve established that temperature and temperature variability is a main driver of their use of spinifex, and behavioural thermoregulation is a pretty messy business. I think this is an interesting idea to explore in the manuscript, certainly. In the abstract, I’d probably avoid such a complex problem myself.
- L28: It’s a nice study, don’t get me wrong, but it’s not quite that novel. I’m pretty sure that Karen Tuff laid the groundwork for this kind of study in 2008, Rachel Garcia published something similar in Proc Roy Soc B. barely a year ago, and even some of my own work has implied the findings for insects (which are ectotherms), although I admit that I didn’t make the same rigorous quantification that you guys did here. Notably none of these forerunners appear to have been identified in this manuscript.

Introduction

- L44: That’s a highly debatable, and I suspect inaccurate statement. This has been, I think, an artefact of data availability and statistical capability. Climatic data have been restricted to large-scale, coarse models for a long time, not least because the ability to measure and model these patterns at fine scales is difficult and computationally expensive, but when even very primitive models have been established at landscape scales and resolutions, they have proven insightful (Ashcroft and Gollan 2012; Tomlinson et al. 2017). Furthermore, the authors lean on an example from the botanical literature to support this, and the botanists are far ahead of zoologists in understanding the importance of physical conditions in structuring niche specialisation at high resolution (Beauregard and de Blois 2014; Thuiller 2013; Tomlinson et al. 2019; Velazco et al. 2017). Species interactions are, of course, critically important at the landscape scale, but I think that this is a naïve and debatable statement to rest an argument on.

- L53: Chicken and egg here. That's true, but foundation species are also often highly constrained to the conditions that they create. So their recruitment and their ability to continue to moderate the abiotic niches of the ecosystem are sometimes tenuous and subject to destabilisation. *Triodia* is actually a really good example (Lewandrowski et al. 2017). I don't disagree with the statement, and this might be one of those points of mine that I warned would be little more than a discussion piece, but this paints a bit of a simplistic image of the place of foundation species in the structure and function of the ecosystem.
- L62: I've not encountered this interpretation before. The association between lizard diversity and ant/termite diversity I have, but not the idea that this is related to spini being spiky. That association isn't intuitive to me. What seems more likely to me is that the spinifex tussocks retain soil moisture and reduce soil temperatures in and around the tussock, and also potentially create softer substrates for ants and termites to dig into, as I think has been shown to drive ant diversity in South African desert ecosystems. I could be wrong, and it's not really a key point to the argument being established here. I get that, but it's a glib simplification of the ecophysiological drivers of niches at fine scales, which is what this manuscript is all about.
- L65: Sorry, what exactly do you mean by "habitat structure" here? Foundation species make an integral contribution to habitat structure, as well as being subject to their own contributions to it. Your argument seems kind of circular.
- L85: These two sentences feel like they should be the opening justification for this paragraph, not the closer. They don't intrinsically lead into the methods as effectively as the previous sentence describing what you proposed to do.

Methods

- L126: The timing of the study seems suboptimal given that one of the main drivers was expected to be temperature. I'm not familiar with the region in NSW, but I'd expect the effect of temperatures on lizards to be most extreme either in February/March or July/August, and that measuring the conditions in those periods would maximise the signal to noise ratio. I know other considerations interfere with the timing of studies like this, but it's a point to consider later in the discussion.
- L136: So just the extremes of high and low temperature, standardised relative to the 18day measurement period, but not standardised in any way relative to local climate or season? No measure of isothermality? No transformation of the temperature data relative to the thermal preferences of the focal species? Fair enough and unavoidable, but points to be specifically considered later in the discussion
- L138: The use of a hurdle model implies that you had zero-inflated data. I suspect that this results from conspiracy between the timing your study and the internal standardisation to identify extremes. The extremes that you identified aren't really local extremes at all, because you measured temperatures in spring. Typically these kinds of studies would be made during seasonal extremes, such as over the summer. Because October/November is a seasonal interchange I suspect that you got a lot of days in October that didn't exceed the top 5% of measurements that would have resulted from your second measurement period in November, and a reverse pattern for your "extreme" cold temperatures. I raise this as a point because you set out to quantify the niche of two spinifex specialists and to investigate the importance of thermal biology in that niche, but this element of your method, and the

internal standardisations that you have made, aren't the best way to do that. I think you should look into ways to standardise to a more appropriate and longer term data set if you can. Maybe look into using the micro_global algorithm: I've used that before in conjunction with temperature logger data and it is remarkably useful (Kearney and Porter 2016; Tomlinson 2020; Tomlinson et al. 2017).

L176: Wait, so the live *Triodia* and the live *Lomandra* were no longer rooted and potentially no longer hydrologically functional? That might have cause quite a difference in the microclimate within the grasses. And also the walls of the mesocosms presumably influenced the irradiation exposure. All unavoidable confounders of your study, I know, but also measurable. You did but data loggers inside each grass clump and somewhere on the surface inside and outside the mesocosm, right?

L180: Was each species placed in it's own separate mesocosm, or were they put in together?

Results

L203: Temperatures ranged from 3.1 to 68.9, but you only report effects for the maximum temperatures. Furthermore, none of the temperatures that you report at the 5% "extremes" falls into a range that is threatening to reptiles. In fact for these lizards, the extreme maxima that you report are short of their preferred temperatures and operational optima (Tomlinson (2019), Rezende and Bozinovic (2019) and references therein). I'd suggest that in the season that you measured temperatures the imperative for the lizards is not overheating during the day (which is why your spinifex specialists didn't use spinifex), but is in fact staying as warm as they can during the night, and maintaining the minimum time to return to operational temperatures following sunrise. You didn't measure their microhabitat preference at night, when you may, or may not have found that the lizards were spinifex specialists after all, but you did measure temperatures at night. Those data will prove insightful.

L220: What was the variance in the dimensions of the spinifex? Did you just have a very variable sample? Or did you just have a very uniform sample of small hummocks?

L236: Spinifex size again. See above

L257: The use of bare ground by these skinks is about consistent with what you'd expect. At these temperatures they were cold, and trying to push themselves up as close to their operational optima as possible. That's not insightful exactly of what defines their thermal niche

L270: Again, these patterns relative to the more obvious preferences by *C. atlas* aren't surprising either. Agamids tend to have higher operational thresholds than skinks, so they were even further from their preferred temperatures. If you'd put a varanid in there I doubt that you would have seen them take cover at all...although given their unfortunate temperament that would have been difficult to distinguish since they'd have spent all their time trying to escape the mesocosm.

L287: OK, I'm glad that these temperatures were measured. This wasn't clear to me in the methods, but maybe I missed it

L288: I suspect that the loss of distinction between live *Triodia* and *Lomandra* was because neither was hydrologically functional anymore, and you no longer had differences in their stomatal

conductance contributing to higher humidity, and reducing temperatures by localised evaporation.

Discussion

- L293: Did you test the functional role of the *Triodia* in defining the niche of these lizards? Can you really say that's what you did given the methodological lapses, and the patterns that I think will become evident in the data if you explore my ideas for Figure 3 and 4. Given that you didn't make your tests when maximum environmental temperatures were in the range that probably defined the thermal niche, I'd say not. Given that you didn't test the preferences of the lizards against the minimum environmental temperatures that probably define the other end of the thermal niche, again I'd say not.
- L300: "...but importantly, very few of the maximum environmental temperatures that we measured fall within a range that is challenging to the lizards that we studied. We speculate that these patterns will be persistent during warmer seasons when the potential value of *Triodia* as a thermal refuge will be substantially higher"
- L302: There's subtleties around this statement, and it's a lot more complex than that. The Sunday reference isn't really articulating all of those to best effect for the objective of this manuscript. You probably want to dive into Martin and Huey (2008) and Angilletta (2009) as a start
- L305: But neither of them had a preference for spinifex at all. They both had an overwhelming preference for bare ground. Given the kind of environmental temperatures that you were exposing these lizards to, this isn't going to tell you much about their preferences for refuge sites.
- L309: Did you work out the actual volumetric and thermal transmissivity constants associated with these differences? I think you'll find them utterly negligible.
- L313: Actually, given the asymmetrical nature of the unimodal thermal performance curve, it seems to me like that's entirely possible. 3 degrees can be the difference between operating at optimal performance and being most of the way to dead once you've exceeded the peak of the performance curve. Your more pressing concern, however, is that most of your measurements were made on the long slow slope below the performance optimum. I think that your lizards were spending most of their time desperately trying to warm up
- L323: Or it's possible that, given their lower thermal preference, and presumably lower T_{ucrit} , climbing into the upper reaches of a live spinifex allows *C. atlas* to escape the boundary layer, potentially catching any air movement to regulate its body by convection, and to capitilise on the increased thermal latency provided by evaporating water vapour from the spinifex leaves, which is probably going to be persistent, because spini rarely closes its stomates entirely. *Ctenophorus spinodomus*, on the other hand, with its higher thermal tolerance and more active foraging strategy, can probably regulate its temperature most effectively at a point closer to the performance optimum by shuttling between exposed habitats and the shade of a *Triodia* hummock, in which case all that matters is the shade, not the extra height to escape the boundary layer. This would really be nailed if you considered the problem from the perspective of performance breadth as well, but there may not be any data describing that.

I'm not disagreeing with your point here, but you started a story where you went looking for *thermal* parameters that may define different realised niches of these lizards, and you're carrying your story on the fact that these are the strongest patterns in your data. We're both telling fairy stories here because the actual patterns in your data are a little inconclusive, although I think that there are ways to get more out of them. I do feel like my fairy story plays closer to the data you've presented, though, rather than falling back on the truism that lizards niche partition, and that's probably what we're seeing here. After all, differential niche partitioning and competitive exclusion would require a cost to either or both parties of sharing a *Triodia* (live or dead) to thermoregulate. This gets at the problem that I identified at the start of this review: these aren't resources in the traditional sense. It's not like the way that *Ctenophorus* eats ants because they have better renal functional than *Ctenopus* and can deal with the salt load better, and the niche partitioning is driven by the fact that if they both ate the same things, then there wouldn't be enough spiders to go around. Individuals of both species could use the same spinifex tussock to thermoregulate at minimal cost to each other. The tussock and the shade aren't being consumed. I'm not seeing a convincing mechanism to underpin the "niche partitioning because lizards do that" argument.

L327: I'm also not buying this line about intraspecific behavioural flexibility. The temperatures that your lizards were exposed to were so low that they hadn't been stressed to the point where behavioural cooling was required. In fact, I think that if you plot your habitat preferences by temperature and put that in the context of a thermal performance curve (see my suggestions for Figure 3), you'll see that most of your lizards were trying to warm up most of the time. I suspect you'll find that the "almost total use" happened when the lizards were at their hottest and were seeking refuge, and I doubt that this happened very often, at least on the basis of what I can see in your temperature data in Figure 2 and Figure 3. I'm also curious, there seems to be a missing chunk of habitat use by *C. atlas* that's not so obvious for *C. spinodomus*. Is this likely to represent the proportion of times that you didn't know where the lizard was? If it does, I next want to know how often did you find *C. atlas* buried under the surface of the sand at the end of the trials? I'm betting that you never saw *C. spinodomus* burrow, but I wonder if *C. atlas* sought thermal refuge below the sand surface, the way that, in my experience, they sometimes do in pit traps.

To cut this diatribe short: I think you're seeing perfectly expected patterns of behavioural thermoregulation here, and you're wrongly ascribing them to something bigger, more elaborate and less likely because you haven't interrogated your data for patterns of thermoregulation. My feeling is that you can do that, and when you do the story might change.

L338: I'm sorry, but from this point on you've entirely lost any sense of plausibility from me. There are simpler explanations for the patterns that you're seeing, that are tied to established physiological mechanisms that provide robust links between pattern and process. Any time you need to start looking for support by drawing parallels between the data you collected for lizard thermal biology and the responses of dolphins to tourist boats, burrowing owls, cattle ranchers or the invasive potential of an evolving mosquito you have to admit you're right at the edges of a rational explanation. Not impossible, but highly unlikely and in no way born out by your data.

L354: And again, you're spinning a fairy story that's much less plausible than the simple, robust, ecophysiological mechanisms. First question about your social signalling hypothesis: were the lizards in the mesocosms alone? I think that they were, so they probably weren't in a

position to be signalling to any other lizard anyway. Lizards tend to start signalling when they actually see another lizard, so without that stimulus, this is a little unlikely. Secondly, Dr Google tells me that both your lizards breed in spring, so that's kind of possible, but you didn't present any evidence that they were in breeding condition. Or, as previously suggested, the lizards were mostly cold, and they were mostly looking to behaviourally thermoregulate by basking in the open.

- L360: It's possible that there really is no actual link between these species and spinifex, and it's possible that the association is just coincidental co-evolution. But, you started your manuscript talking about the critical value of foundational species, and how spinifex is a critical foundational species in arid Australia. Unless you've totally abandoned that line, it seems to me more plausible that your lizards were not yet physiologically stressed to the point that their dependence on the spinifex tussocks became apparent. Furthermore, you didn't offer very large or very dense tussocks to the lizards in your mesocosm trial, so the thermal benefit that they provided, even in the small number of cases where the lizards apparently did enter thermal stress and seek refuge, may have been minimal. Instead of seeking spurious potential support for the patterns you identified in these elaborate "what if" stories, you could just look to the design of your experiment with a slightly more robust understanding of thermal biology and state that the confounding factors of your design somewhat limit your interpretations and your capacity to test your hypotheses.
- L376: Also, both your species are active foragers, willing to roam over relatively large areas (I don't know what the home range of the lizards is), and to forage actively in a number of detailed microhabitats. They're not a sit-and-wait predator the way that a pit trap is, so there may be subtleties in their foraging preferences that aren't detected by bulk collection. Although to be honest, I agree with you, I doubt that insect abundance is what's driving this.
- L391: This is also a bit of a far-fetched story, and there's a much simpler one that is also much more robustly established in the literature. There is a cost to an ectotherm seeking refuge from predators in a cool microhabitat when they are themselves a long way below their operational T_{opt} . They get cold, and then it costs them time and foraging opportunity to warm up again. Lizards won't seek refuge, and will spend less time in refuges if doing so imposes a substantial thermal cost (i.e. when they are a long way from T_{opt} in the way that your lizards were). See Cooper Jr and Wilson (2008) and Polo et al. (2005). In the light of that pretty robust, simple and mechanistic example that also fits with the context of your data, I'm not buying any of the speculations in this paragraph.
- L402: Your conclusions need work. They're consistently speculative and sort of implausible. They also suggest an array of future work, large elements of which have been undertaken in analogous habitats or were explored quite some time ago. Most importantly, your conclusions aren't conclusions about your work at all. There's no take-away message here about what you found, there's just a series of speculations about what other studies might be done to provide support for some of the edgier ideas that you've explored in the discussion. Firstly, most of those discussion points become superfluous when more robust theoretically-rooted explanations are deployed. Secondly, the conclusion is supposed to have some conclusions and take-home messages, but I think you're struggling because you don't really know what your data are showing you. I really hope that some of the ideas I've put down for you here provide you with a bit more insight into what your data are actually showing, and a bit more of a theoretical context for your interpretation. You actually have some room to play with this, but you need to start looking at your data in the right way.

Figures and Tables

- Fig 1: There is reference to the spatial configuration of the three enclosures, and a treatment rotation that isn't articulated in the methods. I'm curious about the importance of this rotation, since from the perspective of the lizard inside the mesocosm they are all identical. Unless the authors are suggesting some kind of magnetosensitivity contributing to the habitat selection by the lizards, which they don't suggest in the methods, I'm not clear on why this matters.
- Fig 2: Mean temperatures aren't really informative in understanding the thermal niche of a species. I'd replot this showing the box and whisker of your extremes. I know that those data are on this plot, and I can see the patterns that I want to know about. I just think it will prove your point better if you plot the data in the most convincing way. In panel b), surely duration shouldn't be represented as a line. There is no continuity between the categorical variables. These should both be represented as columns or boxes.
- Fig3/4: These two don't really make the best use of the data that you have. Fair call that you've shown in Fig 3 that the lizards have a preference for bare ground, but so what? It doesn't demonstrate a causal link with temperature. If you instead plotted the proportion of time spent in habitats of different temperatures, then you have a demonstrated causal link, especially if you take the temperature from your logger at the time that the photo was captured. I think you can do this if I understand the design of your experiment properly. You could even colour code the points in your plot to show that different habitats are associated with different temperatures, all on the one plot.

Now if you want to take that one step further, and establish not just a causal link, but a *mechanistic* link, then you want to know where on the thermal performance curve your lizards are sitting at that temperature. I'm pretty sure that there's good thermal performance data for *C. atlas*, but I'm not sure about *C. spinodomus*. That said, I think I published a curve for a congeneric *Ctenophorus* in my 2019 paper, and Rezende and Bozinovic (2019) published a curve for a *Ctenotus*. The point is that you could access these data and redraft the figure showing time spent by the lizards at different temperatures, but instead of colour coding the points by your four vegetation types, you could code them by reference to the thermal optimum (i.e. 0-90% of thermal opt: basking/thermoregulating; 90-100% of thermal opt: active/foraging; 100-110% of thermal opt: refuge seeking; >110% of thermal opt: dying rapidly). I suspect you'll then find that most of your lizards spent most of their time desperately thermoregulating, and really what you wanted to know for this season was how much time did they spend overnight minimising the distance from their thermal optimum, but you're never going to know that unless you try what I'm suggesting.

Further Reading

- Angilletta, M.J.J., 2009. Thermal Adaptation: a Theoretical and Empirical Synthesis. Oxford University Press, Oxford.
- Ashcroft, M.B., Gollan, J.R., 2012. Fine-resolution (25 m) topoclimatic grids of near-surface (5 cm) extreme temperatures and humidities across various habitats in a large (200 × 300 km) and diverse region. *International Journal of Climatology* 32, 2134–2148.
- Beauregard, F., de Blois, S., 2014. Beyond a climate-centric view of plant distribution: edaphic variables add value to distribution models. *PLoS ONE* 9, e92642.
- Cooper Jr, W.E., Wilson, D.S., 2008. Thermal cost of refuge use affects refuge entry and hiding time by striped plateau lizards *Sceloporus virgatus*. *Herpetologica* 64, 406-412.

- Kearney, M.R., Porter, W.P., 2016. NicheMapR-an R package for biophysical modelling: the microclimate model. *Ecography* doi: [10.1111/ecog.02360].
- Lewandowski, W., Erickson, T.E., Dixon, K.W., Stevens, J.C., 2017. Increasing the germination envelope under water stress improves seedling emergence in two dominant grass species across different pulse rainfall events. *Journal of Applied Ecology* 54, 997-1007.
- Martin, T.L., Huey, R.B., 2008. Why "suboptimal" is optimal: Jensen's inequality and ectotherm thermal preferences. *The American Naturalist* 171, E102-E118.
- Polo, V., López, P., Martín, J., 2005. Balancing the thermal costs and benefits of refuge use to cope with persistent attacks from predators: a model and an experiment with an alpine lizard. *Evolutionary Ecology Research* 7, 23-35.
- Rezende, E.L., Bozinovic, F., 2019. Thermal performance across levels of biological organization. *Philosophical Transactions of the Royal Society B* 374, 20180549.
- Thuiller, W., 2013. On the importance of edaphic variables to predict plant species distributions—limits and prospects. *Journal of Vegetation Science* 24, 591-592.
- Tomlinson, S., 2019. The mathematics of thermal sub-optimality: Nonlinear regression approaches to thermal performance in reptile metabolic rates. *Journal of Thermal Biology* IN PRESS.
- Tomlinson, S., 2020. The construction of small-scale, quasi-mechanistic spatial models of insect energetics in habitat restoration: a case study of beetles in Western Australia. *Diversity and Distributions* IN PRESS.
- Tomlinson, S., Lewandowski, W., Elliott, C.P., Miller, B.P., Turner, S.R., 2019. High resolution distribution modelling of a threatened short-range endemic plant informed by edaphic factors. *Ecology and Evolution* AT REVIEW.
- Tomlinson, S., Webber, B.L., Bradshaw, S.D., Dixon, K.W., Renton, M., 2017. Incorporating biophysical ecology into high-resolution restoration targets: insect pollinator habitat suitability models. *Restoration Ecology* 26, 338-347.
- Velazco, S.J.E., Galvão, F., Villalobos, F., De Marco Júnior, P., 2017. Using worldwide edaphic data to model plant species niches: An assessment at a continental extent. *PLoS ONE* 12, e0186025.

Appendix B

Response to referees' comments

Referee 1

Abstract

1. L14: "Foundation" species is a term I've not encountered before. I get what it is, it's clearly analogous to what I think of as "keystone" species. I don't really play in the pure ecology literature much, but is there a specific reason for changing the term? I'm honestly just curious
Response: The terms are indeed similar and both relate to highly important species, but keystone species are often higher order organisms that influence other species disproportionately to their often comparatively low abundance. Foundation species usually occupy lower trophic levels and are typically abundant, and hence are the appropriate classification for *Triodia*.
2. L18: "Resources" is a particularly loose term for what you looked at, and somewhat misleading in the context. Temperature is not a resource, it's a physical attribute of the microclimate. Similarly, predation risk isn't exactly a "resource" either. Given that one of the major drivers of microhabitat selection by ectotherms is thermo-energetic, when you say "resources", I immediately expect something measured in Joules. Maybe "attributes" or something similar is a better term
Response: We agree and have changed 'resources' to 'attributes'.
3. L20: Maybe identify your species here. It's a bit of a cliffhanger waiting for 2 lines to see what they actually were
Response: We have included the species names earlier, at L19.
4. L25: Dead spinifex with increasing temperatures? Spinifex doesn't have temperatures. I think you mean that it preferred dead spinifex, especially at warmer air temperatures
Response: We have amended L24 to '..., particularly at warmer air temperatures'.
5. L26: What three plant types? Live spinifex, dead spinifex and *Lomandra*, or spinifex, *Lomandra* and bare ground? Or live spinifex, dead spinifex, *Lomandra* or bare ground?
Response: We have removed reference to three plant types and changed L25 to 'available microhabitats'.
6. L26: Variation in individual preference is an interesting finding. It's impossible to unpack in an Abstract, but I think it's too simplistic to say "spinifex specialists should use spinifex" when you've established that temperature and temperature variability is a main driver of their use of spinifex, and behavioural thermoregulation is a pretty messy business. I think this is an interesting idea to explore in the manuscript, certainly. In the abstract, I'd probably avoid such a complex problem myself.
Response: We have removed 'despite the species being considered spinifex specialists'.
7. L28: It's a nice study, don't get me wrong, but it's not quite that novel. I'm pretty sure that Karen Tuff laid the groundwork for this kind of study in 2008, Rachel Garcia published something similar in Proc Roy Soc B. barely a year ago, and even some of my own work has implied the findings for insects (which are ectotherms), although I admit that I didn't make the same rigorous quantification that you guys did here. Notably none of these forerunners appear to have been identified in this manuscript.

8. Response: We have removed mention of ‘novel’ and now reference Tuff (2016) and Garcia (2019) at L52 and L406 respectively.

Introduction

9. L44: That’s a highly debatable, and I suspect inaccurate statement. This has been, I think, an artefact of data availability and statistical capability. Climatic data have been restricted to large-scale, coarse models for a long time, not least because the ability to measure and model these patterns at fine scales is difficult and computationally expensive, but when even very primitive models have been established at landscape scales and resolutions, they have proven insightful (Ashcroft and Gollan 2012; Tomlinson et al. 2017). Furthermore, the authors lean on an example from the botanical literature to support this, and the botanists are far ahead of zoologists in understanding the importance of physical conditions in structuring niche specialisation at high resolution (Beauregard and de Blois 2014; Thuiller 2013; Tomlinson et al. 2019; Velazco et al. 2017). Species interactions are, of course, critically important at the landscape scale, but I think that this is a naïve and debatable statement to rest an argument on.

Response: These patterns are well supported by the literature, but we agree this may nevertheless be an artefact of limitations in scientific techniques. We have therefore removed the following ‘with climatic variables dominating at large scales, and species interactions more important at smaller scales’ to avoid any ambiguity.

10. L53: Chicken and egg here. That’s true, but foundation species are also often highly constrained to the conditions that they create. So their recruitment and their ability to continue to moderate the abiotic niches of the ecosystem are sometimes tenuous and subject to destabilisation. *Triodia* is actually a really good example (Lewandowski et al. 2017). I don’t disagree with the statement, and this might be one of those points of mine that I warned would be little more than a discussion piece, but this paints a bit of a simplistic image of the place of foundation species in the structure and function of the ecosystem.

Response: We agree that the statement is simplistic, particularly regarding *Triodia*. However, given the general, broad context within which we mention this near the start of the introduction we feel it is suitable in its current format.

11. L62: I’ve not encountered this interpretation before. The association between lizard diversity and ant/termite diversity I have, but not the idea that this is related to spini being spiky. That association isn’t intuitive to me. What seems more likely to me is that the spinifex tussocks retain soil moisture and reduce soil temperatures in and around the tussock, and also potentially create softer substrates for ants and termites to dig into, as I think has been shown to drive ant diversity in South African desert ecosystems. I could be wrong, and it’s not really a key point to the argument being established here. I get that, but it’s a glib simplification of the ecophysiological drivers of niches at fine scales, which is what this manuscript is all about.

Response: We agree this sentence is misleading and have generalised the mechanism behind the association by removing reference to ‘spiky’.

12. L65: Sorry, what exactly do you mean by “habitat structure” here? Foundation species make an integral contribution to habitat structure, as well as being subject to their own contributions to it. Your argument seems kind of circular.

Response: We agree there is a circular issue. Due to the inherent complications alluded to here, much of the literature does not differentiate biotic/abiotic impacts of vegetation from purely physical structural effects. We have amended this text as follows, including changing ‘habitat structure’ to ‘vegetation structure’, L62 ‘The inter-connected nature of vegetation

structure and biotic and abiotic factors makes it challenging to determine precisely how foundation species influence other organisms. For example, the physical form of *Triodia* hummocks may alter temperature regimes and substrate properties, facilitating its use by burrowing animals, which in turn further alter habitat properties.’

13. L85: These two sentences feel like they should be the opening justification for this paragraph, not the closer. They don’t intrinsically lead into the methods as effectively as the previous sentence describing what you proposed to do.

Response: We agree and have repositioned the two sentences to the start of the paragraph.

Methods

14. L126: The timing of the study seems suboptimal given that one of the main drivers was expected to be temperature. I’m not familiar with the region in NSW, but I’d expect the effect of temperatures on lizards to be most extreme either in February/March or July/August, and that measuring the conditions in those periods would maximise the signal to noise ratio. I know other considerations interfere with the timing of studies like this, but it’s a point to consider later in the discussion.

Response: We agree that measuring temperatures over additional months would be optimal and we have added this consideration to the discussion at L305 ‘Interestingly, the size of spinifex clumps did not appear to influence these thermal patterns. As we conducted our temperature surveys in spring, we speculate that patterns of average and extreme temperatures would be more pronounced during summer and winter, when the potential value of *Triodia* as a thermal refuge may be substantially higher’.

15. L136: So just the extremes of high and low temperature, standardised relative to the 18 day measurement period, but not standardised in any way relative to local climate or season? No measure of isothermality? No transformation of the temperature data relative to the thermal preferences of the focal species? Fair enough and unavoidable, but points to be specifically considered later in the discussion.

Response: We have now adjusted our definition of temperature extremes (see response to Comment 16) and have added consideration of the spring measurement window to the discussion (see response to Comment 14).

16. L138: The use of a hurdle model implies that you had zero-inflated data. I suspect that this results from conspiracy between the timing your study and the internal standardisation to identify extremes. The extremes that you identified aren’t really local extremes at all, because you measured temperatures in spring. Typically these kinds of studies would be made during seasonal extremes, such as over the summer. Because October/November is a seasonal interchange I suspect that you got a lot of days in October that didn’t exceed the top 5% of measurements that would have resulted from your second measurement period in November, and a reverse pattern for your “extreme” cold temperatures. I raise this as a point because you set out to quantify the niche of two spinifex specialists and to investigate the importance of thermal biology in that niche, but this element of your method, and the internal standardisations that you have made, aren’t the best way to do that. I think you should look into ways to standardise to a more appropriate and longer term data set if you can. Maybe look into using the micro-global algorithm: I’ve used that before in conjunction with temperature logger data and it is remarkably useful (Kearney and Porter 2016; Tomlinson 2020; Tomlinson et al. 2017).

Response: We used a hurdle model to inform us of both the frequency and duration of ‘extreme’ events within a single model. As we are looking at extreme values, by definition we would expect an abundance of ‘zeroes’ representing the many non-extreme temperature

values. While we therefore believe our modelling approach is sound, we accept that our standardisation technique is limited. While standardizing data to a longer term data set and employing a micro-global algorithm has benefits, it also comes with some notable limitations (i.e. assumptions used in extrapolation). As such we feel using our raw, measured data and temperature thresholds that are ecologically relevant to the thermal biology of these lizards is more intuitive and appropriate to address our key aims.

Further, we recorded a large range of temperatures (3.1°C to 68.9°C) and found significant and logical differences across treatments, suggesting our approach was suitable. To reflect our shift from arbitrary to ecologically relevant ‘extremes’ we have updated Methods at L125 ‘To examine the effects of physiologically challenging temperatures (hereafter referred to as ‘extreme’) for our study species we used an upper threshold of 45°C and a lower threshold of 9.7°C. These temperatures represent the approximate critical thermal maximum (CT_{max}) of *C. spinodomus* and two congeneric species of *Ctenotus* (*Ctenotus regius* and *Ctenotus uber*) with overlapping distributions to *C. atlas*, and the average critical thermal minimum (CT_{min}) temperature of *C. regius* and *C. uber* respectively (29,44). We were unable to find suitable data on the CT_{min} for *C. spinodomus*, nor other sympatric *Ctenophorus* spp., thus we used the same value as for *C. atlas*’. We have also updated the Results at L225 and Figures 2a and 2b.

17. L176: Wait, so the live *Triodia* and the live *Lomandra* were no longer rooted and potentially no longer hydrologically functional? That might have caused quite a difference in the microclimate within the grasses. And also the walls of the mesocosms presumably influenced the irradiation exposure. All unavoidable confounders of your study, I know, but also measurable. You did put data loggers inside each grass clump and somewhere on the surface inside and outside the mesocosm, right?

Response: We did place temperature loggers in those locations and the data are presented in Appendix Figure 5. We have added a further line to Methods L180: ‘We also placed iButton temperature loggers within treatments inside enclosures to assess the effects of transplanting and the influence of the mesocosms on irradiation exposure.’

18. L180: Was each species placed in it’s own separate mesocosm, or were they put in together?

Response: We mention lizards were individually placed into enclosures at L171 and we now reiterate this at L361.

Results

19. L203: Temperatures ranged from 3.1 to 68.9, but you only report effects for the maximum temperatures. Furthermore, none of the temperatures that you report at the 5% “extremes” falls into a range that is threatening to reptiles. In fact for these lizards, the extreme maxima that you report are short of their preferred temperatures and operational optima (Tomlinson (2019), Rezende and Bozinovic (2019) and references therein). I’d suggest that in the season that you measured temperatures the imperative for the lizards is not overheating during the day (which is why your spinifex specialists didn’t use spinifex), but is in fact staying as warm as they can during the night, and maintaining the minimum time to return to operational temperatures following sunrise. You didn’t measure their microhabitat preference at night, when you may, or may not have found that the lizards were spinifex specialists after all, but you did measure temperatures at night. Those data will prove insightful.

Response: The temperatures we reported were mean daily temperatures. We have now amended this to also report mean daily maximum and minimum temperatures at L215 ‘Analysis of mean daily maximum and minimum temperatures revealed similar patterns,

with bare ground recording significantly cooler minimum daily temperatures (15.9°C, 15.3–16.6, $p < 0.001$) and significantly warmer maximum daily temperatures (43.2°C, 41.6–44.8, $p < 0.001$) than all other treatments. Live spinifex and dead spinifex were statistically indistinguishable from one another ($p = 0.823$) for both minimum (17.5°C, 16.8–18.1) and maximum (live = 31.8°C, 30.2 – 33.5; dead = 31.6°C, 30.0–33.3) daily temperatures. Lomandra only differed statistically from spinifex at maximum daily temperatures (34.1°C, 32.5–35.7, $p < 0.001$).’.

We have also adjusted our classification of ‘extreme’ to better reflect those temperatures that are challenging to these lizards ($9.7 < x < 45^\circ\text{C}$), defining these terms in the Methods, reporting the values and updating Figure 2 and the statistical results (see response to Comment 16).

Finally, we agree that it would be beneficial to examine night-time refuge selection and have added the following at L397 ‘In addition, extending mesocosm trials through the night may provide greater insight into refuge preferences when daily temperatures are likely to be at their coolest, and behaviour associated with foraging is substantially reduced or eliminated’.

20. L220: What was the variance in the dimensions of the spinifex? Did you just have a very variable sample? Or did you just have a very uniform sample of small hummocks?

Response: We have added summary statistics of clump dimensions at L223 ‘Spinifex clump sizes within treatments spanned a height range from 11 cm to 98 cm (mean live = 44.3 cm, s.d. = 11.8; mean dead = 32.0 cm, s.d. = 11.9), and widths ranged from 45 cm to 199 cm (mean live = 71.6 cm, s.d. = 24.6; mean dead = 78 cm, s.e. = 33.9).’.

21. L236: Spinifex size again. See above

Response: Please see response to Comment 20.

22. L257: The use of bare ground by these skinks is about consistent with what you’d expect. At these temperatures they were cold, and trying to push themselves up as close to their operational optima as possible. That’s not insightful exactly of what defines their thermal niche

Response: We have revised Figure 3 and include more detailed reporting of the temperatures at which study lizards were exposed at L261 ‘There were 5,468 observations (9.9% of 55,086 total observations) for the skink *C. atlas* where temperatures exceeded the CT_{\max} threshold ($> 45^\circ\text{C}$). Similarly, 7,732 observations (8.3% of 92,775 total observations) exceeded the CT_{\max} for the dragon *C. spinodomus*. A further 13,857 observations (14.9% of total) occurred below the voluntary minimum (20°C) for *C. spinodomus* (Figure 3)’.

While the majority of temperatures were within the normal active zone, a large number of data points occurred at challenging/threatening temperatures for these species. Further, our study was an attempt to ascertain which mechanism (thermal, predation, food) was most important. As such we wanted a considerable amount of data across the thermal performance curve to include behaviour not just where lizards were challenged thermally but also where they were comfortable and active. We feel such data are important and informative about which microhabitats are preferred at what times (as illustrated through Figure 4 and the model outputs).

We also acknowledge this limitation in the Discussion at L347 ‘This was most likely due to basking, as reptiles often select relatively warm open microhabitat. particularly when temperatures are below thermal optimums (61–63). However, our ability to interpret this

strong preference for open microhabitat is limited by the relatively low proportion of time at which lizards were subject to thermal stress’.

23. L270: Again, these patterns relative to the more obvious preferences by *C. atlas* aren’t surprising either. Agamids tend to have higher operational thresholds than skinks, so they were even further from their preferred temperatures. If you’d put a varanid in there I doubt that you would have seen them take cover at all...although given their unfortunate temperament that would have been difficult to distinguish since they’d have spent all their time trying to escape the mesocosm.

Response: We have now more clearly reported the temperatures to which lizards were exposed during trials and we agree basking behaviour is a likely explanation for a significant portion of the bare ground usage we observed (see Comment 22).

24. L288: I suspect that the loss of distinction between live *Triodia* and *Lomandra* was because neither was hydrologically functional anymore, and you no longer had differences in their stomatal conductance contributing to higher humidity, and reducing temperatures by localised evaporation.

Response: This is a good point and we have added this consideration to the Discussion at L355 ‘Further, the alteration of hydrological function from transplanting the plants may have reduced differences in stomatal conductance, humidity and subsequently temperature differences through evaporation’.

We also added the following information to the Methods at L175 ‘We retained as much of the root clump of plants as possible, but removed excess retained soil by hand prior to positioning within mesocosms’.

Discussion

25. L293: Did you test the functional role of the *Triodia* in defining the niche of these lizards? Can you really say that’s what you did given the methodological lapses, and the patterns that I think will become evident in the data if you explore my ideas for Figure 3 and 4. Given that you didn’t make your tests when maximum environmental temperatures were in the range that probably defined the thermal niche, I’d say not. Given that you didn’t test the preferences of the lizards against the minimum environmental temperatures that probably define the other end of the thermal niche, again I’d say not.

Response: We believe we did test the functional environment of spinifex for two lizards though there were limitations and we now acknowledge these in the Discussion based on the feedback. We have revised the wording to remove ambiguity and follow the definitions outlined by Kearney (2006) at L296 ‘We tested the functional environment of a foundation species (spinifex grass *T. scariosa*) for two putative spinifex specialist lizard species’.

While we do not claim to have defined the thermal niche, the evidence we present firmly points to the dominant role of thermal benefits over food or refuge, as a driving function of this plant. Further, our data includes a large number of observations where temperatures were outside their voluntary minimum and/or CT_{max} , which we feel represents a robust insight into lizard behaviour at challenging temperatures.

26. L300: “...but importantly, very few of the maximum environmental temperatures that we measured fall within a range that is challenging to the lizards that we studied. We speculate that these patterns will be persistent during warmer seasons when the potential value of *Triodia* as a thermal refuge will be substantially higher”

Response: We have clarified our reporting of maximum daily temperatures (see response to Comment 22) to show that reasonable numbers of temperatures could be considered

challenging, though we acknowledge that the majority of temperatures were not (see responses to Comments 34 and 14).

27. L302: There's subtleties around this statement, and it's a lot more complex than that. The Sunday reference isn't really articulating all of those to best effect for the objective of this manuscript. You probably want to dive into Martin and Huey (2008) and Angilletta (2009) as a start.

Response: We now include both recommended references and have rephrased the sentence at L309 'Terrestrial ectotherms have limited physiological thermal tolerance, and asymmetric temperature-fitness curves mean fitness is depressed more rapidly at body temperatures above optimal levels (53,54)'.

28. L305: But neither of them had a preference for spinifex at all. They both had an overwhelming preference for bare ground. Given the kind of environmental temperatures that you were exposing these lizards to, this isn't going to tell you much about their preferences for refuge sites.

Response: We report the relative rate of use of live vs dead spinifex here, which was aligned with our main questions. The pattern of predominant use of bare ground is discussed at L346 (also see response to Comment 34). We now address the temperatures at which these lizards were exposed in more detail (see response to Comment 22). While bare ground was preferred over any single refuge type, we recorded ~67,000 observations of lizards using refuges as well as 13,200 observations that exceeded CT thresholds so we feel this pattern is noteworthy.

29. L309: Did you work out the actual volumetric and thermal transmissivity constants associated with these differences? I think you'll find them utterly negligible.

Response: We agree and have removed the following text '*Ctenotus atlas* are larger (average SVL 59.3 mm, s.e. 1.92) and darker (Figure 1) than *C. spinodomus* (45.7 mm, 0.54) and increased body size can result in lower critical thermal maxima in lizards (50).'

30. L313: Actually, given the asymmetrical nature of the unimodal thermal performance curve, it seems to me like that's entirely possible. 3 degrees can be the difference between operating at optimal performance and being most of the way to dead once you've exceeded the peak of the performance curve. Your more pressing concern, however, is that most of your measurements were made on the long slow slope below the performance optimum. I think that your lizards were spending most of their time desperately trying to warm up

Response: We have added more detail on the temperatures to which lizards were exposed and our data captures a large number of observations at both ends of the thermal performance curve (see Comment 22 and Figure 3). The preference of *C. atlas* for dead spinifex as temperatures increase, including temperatures that exceed their CT_{max} , despite preferring lower temperatures than *C. spinodomus* seems to require explanations beyond purely thermal mechanisms (also see response to Comment 31).

31. L323: Or it's possible that, given their lower thermal preference, and presumably lower Tucrit, climbing into the upper reaches of a live spinifex allows *C. atlas* to escape the boundary layer, potentially catching any air movement to regulate its body by convection, and to capitalise on the increased thermal latency provided by evaporating water vapour from the spinifex leaves, which is probably going to be persistent, because spini rarely closes its stomates entirely. *Ctenophorus spinodomus*, on the other hand, with its higher thermal tolerance and more active foraging strategy, can probably regulate its temperature most effectively at a point closer to the performance optimum by shuttling between exposed

habitats and the shade of a *Triodia* hummock, in which case all that matters is the shade, not the extra height to escape the boundary layer. This would really be nailed if you considered the problem from the perspective of performance breadth as well, but there may not be any data describing that. I'm not disagreeing with your point here, but you started a story where you went looking for *thermal* parameters that may define different realised niches of these lizards, and you're carrying your story on the fact that these are the strongest patterns in your data. We're both telling fairy stories here because the actual patterns in your data are a little inconclusive, although I think that there are ways to get more out of them. I do feel like my fairy story plays closer to the data you've presented, though, rather than falling back on the truism that lizards niche partition, and that's probably what we're seeing here. After all, differential niche partitioning and competitive exclusion would require a cost to either or both parties of sharing a *Triodia* (live or dead) to thermoregulate. This gets at the problem that I identified at the start of this review: these aren't resources in the traditional sense. It's not like the way that *Ctenophorus* eats ants because they have better renal functional than *Ctenotus* and can deal with the salt load better, and the niche partitioning is driven by the fact that if they both ate the same things, then there wouldn't be enough spiders to go around. Individuals of both species could use the same spinifex tussock to thermoregulate at minimal cost to each other. The tussock and the shade aren't being consumed. I'm not seeing a convincing mechanism to underpin the "niche partitioning because lizards do that" argument.

Response: We acknowledge in the manuscript that our idea is speculative (see L324) and we also agree our argument about foraging strategies does not rely on resources being competed over. We have therefore removed mention of competition and niche partitioning. We have also added consideration of evaporative cooling effects (see response to Comment 33).

Escaping the boundary layer is a plausible explanation for why *C. atlas* may be more prone to basking higher up in tussocks (either live or dead), but we did not collect appropriate data to investigate this aspect, nor do we feel this pattern holds in our study system, with *C. spinodomus* appearing at least as likely to use the top of spinifex clumps (based on field observations). Further, *C. atlas* tended to use dead spinifex clumps more often as temperatures rose, which on average achieve lower physical heights and feasibly offer less evaporative cooling than live clumps.

32. L327: I'm also not buying this line about intraspecific behavioural flexibility. The temperatures that your lizards were exposed to were so low that they hadn't been stressed to the point where behavioural cooling was required. In fact, I think that if you plot your habitat preferences by temperature and put that in the context of a thermal performance curve (see my suggestions for Figure 3), you'll see that most of your lizards were trying to warm up most of the time. I suspect you'll find that the "almost total use" happened when the lizards were at their hottest and were seeking refuge, and I doubt that this happened very often, at least on the basis of what I can see in your temperature data in Figure 2 and Figure 3. I'm also curious, there seems to be a missing chunk of habitat use by *C. atlas* that's not so obvious for *C. spinidomus*. Is this likely to represent the proportion of times that you didn't know where the lizard was? If it does, I next want to know how often did you find *C. atlas* buried under the surface of the sand at the end of the trials? I'm betting that you never saw *C. spinodomus* burrow, but I wonder if *C. atlas* sought thermal refuge below the sand surface, the way that, in my experience, they sometimes do in pit traps. To cut this diatribe short: I think you're seeing perfectly expected patterns of behavioural thermoregulation here, and you're wrongly ascribing them to something bigger, more elaborate and less likely

because you haven't interrogated your data for patterns of thermoregulation. My feeling is that you can do that, and when you do the story might change.

Response: We demonstrate the importance of temperature in determining microhabitat preference, with use of bare ground decreasing with increasing temperature (see Results L273 & L283 and Figure 4). We have now added an additional Figure to the Appendix (Figure 3) to establish the variability in use of bare ground versus plants, before looking purely at differences in plant types (Appendix Figure 4). We acknowledge we did not explicitly consider the influence of temperature when interpreting individual variations in response. As such, for both appendix figures, we have now overlaid the minimum, maximum and average temperatures to which individual lizards were exposed during their trials. This shows no clear trend for trials that were conducted at hotter temperatures to record greater use of bare ground, at the individual level. We have also moderated our Discussion of these patterns at L334 'Although some of the overall variation in plant selection was due to temperature, thermal effects did not appear to strongly influence differences between individuals. That is, lizards trialled at the same time (in separate mesocosms), and therefore under comparable environmental conditions, often preferred different plant type'.

Two *C. spinodomus* and four *C. atlas* were recorded burrowing or digging but no such activity was sustained for any significant period and no burrow was completed nor utilised.

33. L338: I'm sorry, but from this point on you've entirely lost any sense of plausibility from me. There are simpler explanations for the patterns that you're seeing, that are tied to established physiological mechanisms that provide robust links between pattern and process. Any time you need to start looking for support by drawing parallels between the data you collected for lizard thermal biology and the responses of dolphins to tourist boats, burrowing owls cattle ranchers or the invasive potential of an evolving mosquito you have to admit you're right at the edges of a rational explanation. Not impossible, but highly unlikely and in no way born out by your data.

Response: This was not directly intended as an explanation for the pattern, but rather an attempt to place our findings in a broader context. Nonetheless, we accept these examples may be misleading and have therefore removed these references.

34. L354: And again, you're spinning a fairy story that's much less plausible than the simple, robust, ecophysiological mechanisms. First question about your social signalling hypothesis: were the lizards in the mesocosms alone? I think that they were, so they probably weren't in a position to be signalling to any other lizard anyway. Lizards tend to start signalling when they actually see another lizard, so without that stimulus, this is a little unlikely. Secondly, Dr Google tells me that both your lizards breed in spring, so that's kind of possible, but you didn't present any evidence that they were in breeding condition. Or, as previously suggested, the lizards were mostly cold, and they were mostly looking to behaviourally thermoregulate by basking in the open.

Response: We have amended the paragraph to temper the importance of these other mechanisms within the context of our mesocosm trials and agree that the ecophysiological explanation is most likely. We have added at L347 'This was most likely due to basking, as reptiles often select relatively warm open microhabitat, particularly when temperatures are below thermal optimums (61–63)' and L359 'Besides thermoregulation, it is possible that courtship display or home-site defence, best performed in conspicuous areas, could partially explain preferences for open microhabitat (64,65). However, sex was not a significant covariate and lizards were tested individually so our results are unlikely to have been influenced by such behaviours'.

35. L360: It's possible that there really is no actual link between these species and spinifex, and it's possible that the association is just coincidental co-evolution. But, you started your manuscript talking about the critical value of foundational species, and how spinifex is a critical foundational species in arid Australia. Unless you've totally abandoned that line, it seems to me more plausible that your lizards were not yet physiologically stressed to the point that their dependence on the spinifex tussocks became apparent. Furthermore, you didn't offer very large or very dense tussocks to the lizards in your mesocosm trial, so the thermal benefit that they provided, even in the small number of cases where the lizards apparently did enter thermal stress and seek refuge, may have been minimal. Instead of seeking spurious potential support for the patterns you identified in these elaborate "what if" stories, you could just look to the design of your experiment with a slightly more robust understanding of thermal biology and state that the confounding factors of your design somewhat limit your interpretations and your capacity to test your hypotheses.
Response: We agree that this is overly speculative and have therefore removed the following text from this paragraph 'Alternatively, because spinifex prefers open, unshaded sites with loose, sandy soil (67), the close association of lizards to spinifex may be over-emphasised by the relatively high prevalence of spinifex in open habitat compared to other plant species. Indeed, the link between lizards and spinifex may to some degree be a more indirect and incidental result of co-evolution, with high lizard diversity within spinifex areas a potential artefact of the prominence of *Triodia* over a vast, contiguous and climatically homogeneous area (27).'
36. L376: Also, both your species are active foragers, willing to roam over relatively large areas (I don't know what the home range of the lizards is), and to forage actively in a number of detailed microhabitats. They're not a sit-and-wait predator the way that a pit trap is, so there may be subtleties in their foraging preferences that aren't detected by bulk collection. Although to be honest, I agree with you, I doubt that insect abundance is what's driving this.
Response: *Ctenophorus spinidomus* have small home ranges and feed exclusively in open ground (Cogger, 1978) but we agree with your point and feel that L370 adequately reflects this uncertainty.
37. L391: This is also a bit of a far-fetched story, and there's a much simpler one that is also much more robustly established in the literature. There is a cost to an ectotherm seeking refuge from predators in a cool microhabitat when they are themselves a long way below their operational T_{opt} . They get cold, and then it costs them time and foraging opportunity to warm up again. Lizards won't seek refuge, and will spend less time in refuges if doing so imposes a substantial thermal cost (i.e. when they are a long way from T_{opt} in the way that your lizards were). See Cooper Jr and Wilson (2008) and Polo et al. (2005). In the light of that pretty robust, simple and mechanistic example that also fits with the context of your data, I'm not buying any of the speculations in this paragraph.
Response: We agree and have entirely removed this paragraph.
38. L402: Your conclusions need work. They're consistently speculative and sort of implausible. They also suggest an array of future work, large elements of which have been undertaken in analogous habitats or were explored quite some time ago. Most importantly, your conclusions aren't conclusions about your work at all. There's no take-away message here about what you found, there's just a series of speculations about what other studies might be done to provide support for some of the edgier ideas that you've explored in the discussion. Firstly, most of those discussion points become superfluous when more robust theoretically-rooted explanations are deployed. Secondly, the conclusion is supposed to have some conclusions and take-home messages, but I think you're struggling because you

don't really know what your data are showing you. I really hope that some of the ideas I've put down for you here provide you with a bit more insight into what your data are actually showing, and a bit more of a theoretical context for your interpretation. You actually have some room to play with this, but you need to start looking at your data in the right way.

Response: As suggested, we have now added a considerable amount of theoretical support and established physiological mechanisms throughout the Discussion, including increased attention to thermal explanations for the patterns we observed (see responses to Comments 32, 33 and 37), and removal of some more speculative elements (see response to Comments 31 and 35). Our stated goal at the start of the paper was to examine the respective merits of three mechanisms driving the use of a foundation species, and we conclude that temperature was likely to be a more important driver of microhabitat use than food or protection for our study species (notwithstanding the acknowledged limitations). We have re-written the concluding paragraph to focus on our key findings and include some broader perspective.

Figures and Tables

39. Fig 1: There is reference to the spatial configuration of the three enclosures, and a treatment rotation that isn't articulated in the methods. I'm curious about the importance of this rotation, since from the perspective of the lizard inside the mesocosm they are all identical. Unless the authors are suggesting some kind of magnetosensitivity contributing to the habitat selection by the lizards, which they don't suggest in the methods, I'm not clear on why this matters.

Response: We have added a sentence to clarify our rationale at L178 'The position of each plant type was switched between the three enclosures to control for any potential effects of slope or shading from the enclosure walls.'

40. Fig 2: Mean temperatures aren't really informative in understanding the thermal niche of a species. I'd replot this showing the box and whisker of your extremes. I know that those data are on this plot, and I can see the patterns that I want to know about. I just think it will prove your point better if you plot the data in the most convincing way. In panel b), surely duration shouldn't be represented as a line. There is no continuity between the categorical variables. These should both be represented as columns or boxes.

Response: We agree and have now presented the information in line with your suggestions, as well as re-defining "extremes" for the purposes of Figure 2b (see Comment 16).

41. Fig3/4: These two don't really make the best use of the data that you have. Fair call that you've shown in Fig 3 that the lizards have a preference for bare ground, but so what? It doesn't demonstrate a causal link with temperature. If you instead plotted the proportion of time spent in habitats of different temperatures, then you have a demonstrated causal link, especially if you take the temperature from your logger at the time that the photo was captured. I think you can do this if I understand the design of your experiment properly. You could even colour code the points in your plot to show that different habitats are associated with different temperatures, all on the one plot. Now if you want to take that one step further, and establish not just a causal link, but a *mechanistic* link, then you want to know where on the thermal performance curve your lizards are sitting at that temperature. I'm pretty sure that there's good thermal performance data for *C. atlas*, but I'm not sure about *C. spinodomus*. That said, I think I published a curve for a congeneric *Ctenophorus* in my 2019 paper, and Rezende and Bozinovic (2019) published a curve for a *Ctenotus*. The point is that you could access these data and redraft the figure showing time spent by the lizards at different temperatures, but instead of colour coding the points by your four vegetation types, you could code them by reference to the thermal optimum (i.e. 0-90% of thermal opt:

basking/thermoregulating; 90- 100% of thermal opt: active/foraging; 100-110% of thermal opt: refuge seeking; >110% of thermal opt: dying rapidly). I suspect you'll then find that most of your lizards spent most of their time desperately thermoregulating, and really what you wanted to know for this season was how much time did they spend overnight minimising the distance from their thermal optimum, but you're never going to know that unless you try what I'm suggesting.

Response: Our efforts to present data according to the recommendations of the reviewer were hindered by differences in the presentation and reporting of thresholds between species within the existing published literature. Nevertheless, we have redrafted Figure 3 (and Appendix Figures 3 and 4) to include the temperatures at which locations of lizards were recorded, along with data density curves to further illustrate the distribution as clearly as possible. We have also overlaid ecologically relevant temperature thresholds. We feel this provides a good context of how microhabitat preferences differ depending on where in their performance curve the lizards are operating. We have retained Figure 4 as it reflects the direct model outputs and we feel provides an effective visual illustration of how preferences for treatment microhabitats differ for the two lizard species as temperatures increase. Namely, use of all microhabitat types drops or remains stable as temperatures increase with the exception of dead spinifex for *C. atlas* and live spinifex for *C. spinodomus*. The decrease in use of bare ground with increasing temperature supports the suggestion regarding basking behaviour.

Referee 2

42. Two small areas might warrant further discussion. One is sex (and reproductive condition) – the wide variation among individuals in habitat preference could, in part, be explained by sex or by whether females are gravid, but that was not mentioned here. Perhaps there was no evidence that sex or reproductive condition drove these differences, but if the authors have the data it might be worth mentioning.

Response: Sex is certainly worthy of investigation and we included sex as a covariate within the mesocosm trials analysis (see Methods L194, Results L274 & L286). However this variable was not significant for either lizard species. We have added further clarification that sex was not a significant covariate within our analyses and that lizards were tested individually, so effects of sexual signalling and mating behaviour were unlikely to be significant (see response to Comment 34).

43. In addition, the 'failure' of the predation experiment using models could be examined more fully. Typically, very, very large numbers of models, over long periods, are required to detect predation (I'm not sure what 4,700 hours means in terms of risk), so there may just have not been enough models. Looking back at the methods you don't actually say how many models there were, that might be good to put in the paper although I suppose it might be in the appendix - I think it should appear in the methods.

Response: We mention a total of 112 models were used in the Methods at L160. We have added the potential benefit of using a greater number of models in the Discussion at L379 'For example, the use of a larger number of models may have enabled us to record more predatory events and potentially reveal differences between treatments'.

44. Given this failure, it might have been nice to see a graph of the different predators you found in the different habitat types (as part of your trapping study, not just on the cameras) – just to get an idea about potential (and also, maybe, perceived) predation in the different habitats. The lizards might respond to expected predation (or perceived predation risk)

rather than actual predation. Further experiments in the mesocosms with predators (in cages, or model predators) could be instructive, and might be worth mentioning.

Response: We mention the potential predators we encountered during trapping at L254, and also the potential for using predatory cues at L389. We have now mentioned the possibility of adding a predatory threat to the mesocosm experiment at L393 'a fully factorial design where multiple parameters of vegetation and predatory stimuli are manipulated may be of most assistance in determining the niche mechanisms that shape use of foundation species. For example, comparing lizard responses to visual and olfactory predatory cues may help determine if spinifex use depends on threat type (e.g. aerial vs ground-based)'. Our vertebrate pitfall arrays were not positioned systematically but rather opportunistically to maximise catches of our study species. Equally, while our invertebrate traps were systematically positioned within treatments, they were (purposefully) too small to catch any vertebrate predators of the lizards. As such we are unable to report on how potential predator abundances varied with habitat type, though we agree this would be useful supporting information.

45. Wordy statements, especially 'considered to be' appear everywhere. If something is 'considered to be' then it just is, unless you are going to show otherwise. So please cut these out

Response: All uses of 'considered' have been removed.

e.g., Lines 91-93 The effect of spinifex on faunal diversity and ecosystem processes has led to it being considered a foundation species. – wordy and awkward sentence – rewrite

Response: We have now re-written this sentence; 'Spinifex is a foundation species due to its effect on faunal diversity and ecosystem processes'.

46. Lines 203-210 – small points but: temperatures differed significantly (rather than just differed – they always differ), and the temperature of bare ground was higher than all the other treatments, and the temperature of spinifex was lower... and they could be differentiated from one another, but were not significantly different.

Response: We have added 'significantly' to L207 and rephrased L211 for improved clarity 'Pairwise comparisons showed that bare ground had significantly higher temperatures than all other treatments ($p < 0.001$) and live and dead spinifex significantly lower temperatures than Lomandra ($p = 0.023$ and 0.010 , respectively), but temperatures between live and dead spinifex were not statistically different from one another ($p = 0.748$).'

47. Line 211: it was not the live spinifex that recorded the lowest number of days...

Response: We have now re-written L225 'Live spinifex had the lowest number of days...'

48. Line 223: You don't need to say what kind of plots they are, start with "mean daily temperatures" etc

Response: 'Boxplots of' removed.

49. Line 391: Burton's legless lizard

Response: Mention of this species has now been removed to address concerns of Referee 1 (see Comment 37).

Appendix C

Kristian Bell, Timothy Doherty and Don Driscoll “Predators, prey or temperature? Mechanisms driving niche use of a foundation plant species by specialist lizards”

Submitted to *Proceedings of the Royal Society B*: RSPB-2020-2633.R1

My recollection of this manuscript is that I liked the idea, but I felt that the data had not been appropriately reported and that several obviously physiological drivers had been overlooked. Looking through the resubmission, I can see a great deal of effort has been made to reshape the arguments and incorporate the suggestions that I have made. It's also now much more apparent to me that some of my initial criticisms were unfair, and resulted from my not seeing the breadth of temperatures under which the lizards had been observed. Overall, I think that these changes and additions have very much clarified the arguments put forward by Bell and colleagues.

Reading the manuscript now, there is much greater and more convincing support for the role of temperature and microclimate driving the patterns reported. There is also a greater acknowledgement of the unavoidable limitations of the study. This presents a much tighter narrative than the first version of the MS, and I recommend that it be accepted after addressing the only typographical error that I noticed (see line by line comments)

Sincerely,

Dr. Sean Tomlinson
Research Fellow

School of Biological Sciences
The University of Adelaide
SA 5005 AUSTRALIA
Email: sean.tomlinson@adelaide.edu.au
www.adelaide.edu.au
CRICOS provider number 00123M

THE UNIVERSITY
of ADELAIDE

Kings Park Science Adjunct
Department of Biodiversity, Conservation and Attractions
WA 6005 AUSTRALIA

 **KINGS PARK | BOLD PARK**
& Botanic Garden

Line-by-line comments

L348: 'microhabitat. particularly' The full stop should be a comma